# Distinct gene regulatory dynamics drive skeletogenic cell fate convergence during vertebrate embryogenesis

Menghan Wang [1,3,6], Ana Di Pietro-Torres[1,6], Christian Feregrino [1,4], Maëva Luxey[1,5], Chloé Moreau[1], Sabrina Fischer[1], Antoine Fages [1], Danilo Ritz [2] & Patrick Tschopp [1] ✉

Cell type repertoires have expanded extensively in metazoan animals, with some clade-specific cells being crucial to evolutionary success. A prime example are the skeletogenic cells of vertebrates. Depending on anatomical location, these cells originate from three different precursor lineages, yet they converge developmentally towards similar cellular phenotypes. Furthermore, their 'skeletogenic competency' arose at distinct evolutionary timepoints, thus questioning to what extent different skeletal body parts rely on truly homologous cell types. Here, we investigate how lineage-specific molecular properties are integrated at the gene regulatory level, to allow for skeletogenic cell fate convergence. Using single-cell functional genomics, we find that distinct transcription factor profiles are inherited from the three precursor states and incorporated at lineage-specific enhancer elements. This lineage-specific regulatory logic suggests that these regionalized skeletogenic cells are distinct cell types, rendering them amenable to individualized selection, to define adaptive morphologies and biomaterial properties in different parts of the vertebrate skeleton.

Metazoan bodies are defined by obligate multicellularity and the presence of functionally, morphologically, and molecularly distinct cell types[1,2]. How to best determine what constitutes a distinct "cell type", however, is still an ongoing debate[1,3–6]. Different molecular metrics have shown promise to delineate cell types across development and evolution, for the repertoire of expressed genes also instructs a cell's form and function. Yet, molecular similarities between cells may occur for different reasons. They can imply homology, i.e., a shared evolutionary history, but can equally result from convergence, drift, concerted evolution, or the co-option of shared gene modules from other developmental contexts[1,7–9]. Additionally, differences in the extracellular signaling environment, or the developmental lineage the cells originate from, further complicate these assessments[5,6,10]. A focus on the underlying regulatory logic that specifies a given cell—from the expressed transcription factors and regulatory RNAs to the resulting protein complexes and the DNA motifs they bind to, up to the level of gene regulatory networks—can help to discriminate between these different scenarios. Accordingly, gene regulatory studies have emerged as a viable experimental approach to help disentangle evolutionary and developmental relationships, both within and across species boundaries, and, thus, resolve potential cell type homologies[1,6,8,11–13].

In this context, vertebrate skeletogenesis offers an opportune model to study the gene regulatory logic of cell fate specification across both developmental and evolutionary timescales. A cartilaginous and often ossified endoskeleton is a hallmark of the vertebrate

[1]Zoology, Department of Environmental Sciences, University of Basel, Basel, Switzerland. [2]Proteomics Core Facility, Biozentrum, University of Basel, Basel, Switzerland. [3]Present address: Department of Biomedicine, University Hospital Basel, University of Basel, Basel, Switzerland. [4]Present address: Max Planck Institute for Molecular Genetics, Berlin, Germany. [5]Present address: MeLis, CNRS UMR 5284, INSERM U1314, Université Claude Bernard Lyon 1, Institut NeuroMyo Gène, Lyon, France. [6]These authors contributed equally: Menghan Wang, Ana Di Pietro-Torres. ✉e-mail: patrick.tschopp@unibas.ch

clade, with many accessory connective tissue types required to build a functional skeleton[14]. Developmentally, the specification of the cells required to build these tissues initiates with mesenchymal precursors (PC) condensing at the onset of vertebrate skeletogenesis and then progressing through distinct, cell type-specific differentiation processes[15–18]. A peculiarity in this process is that—unlike for many other cell lineages (Fig. 1a, left)—these multipotent skeletal progenitors can arise convergently from distinct embryonic sources (Fig. 1a, right). Namely, depending on anatomical location, three distinct embryonic lineages—the cranial neural crest, the somitic sclerotome, and the somatopleure of the lateral plate mesoderm—are the developmental origin of the cranial, axial, and appendicular skeleton, respectively[14]

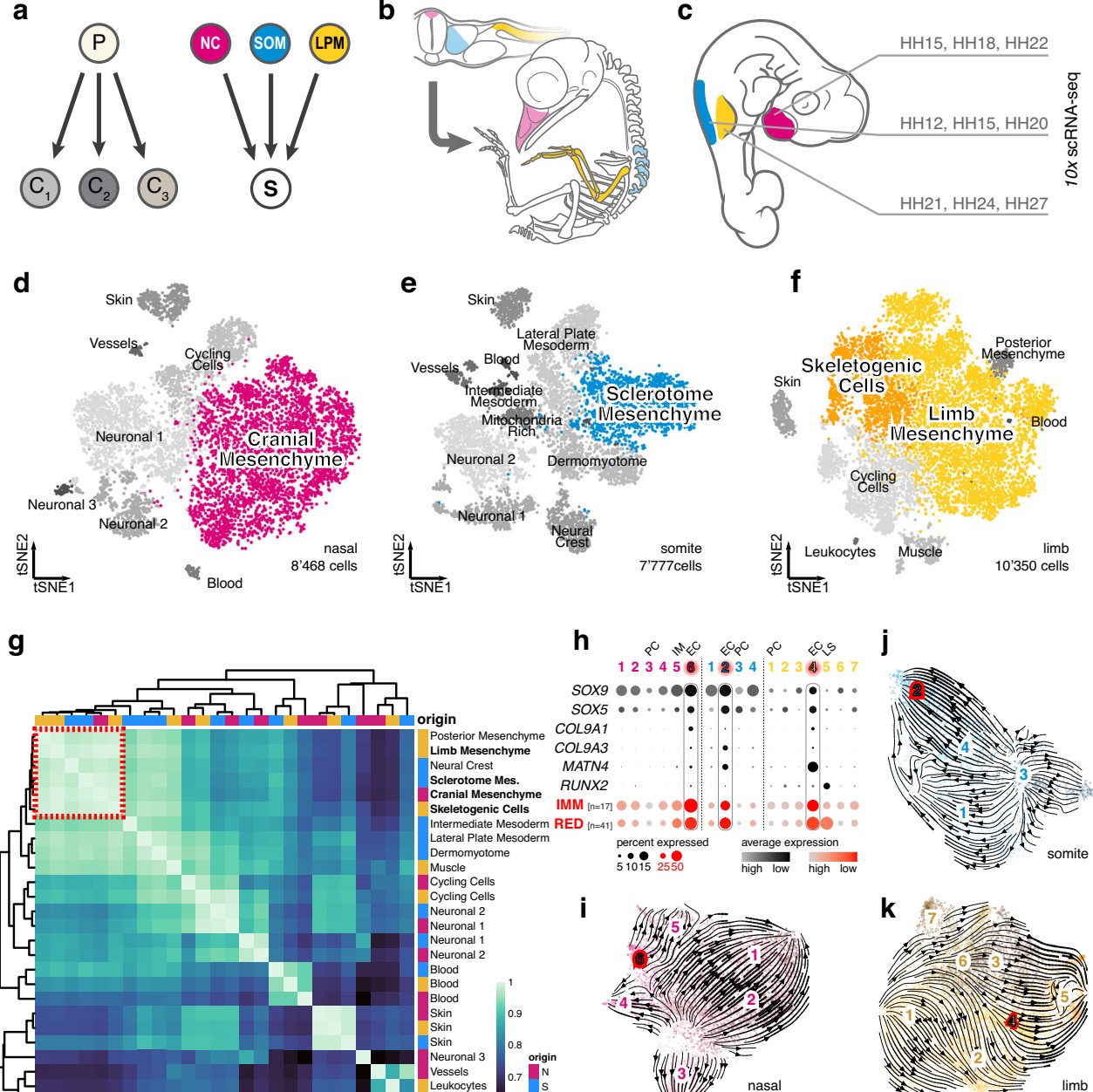

**Fig. 1 | A convergent transcriptomic signature in skeletogenic cells of different embryonic origins. a** Divergent diversifying versus convergent cell fate decisions. Precursor cells (P) usually differentiate into functionally distinct cell types (C$_{1-3}$). During vertebrate skeletogenesis, however, three distinct embryonic lineages— the neural crest (NC, magenta), the somitic mesoderm (SOM, blue), and the lateral plate mesoderm (LPM, yellow)—give rise to functionally similar skeletogenic cells (S). **b** Early embryonic origins and eventual anatomical locations of skeletogenic cells sampled in this study. **c** Sampling scheme to assess the transcriptional dynamics of skeletogenic convergence across anatomical locations (color-coded) and embryonic stages (Hamburger–Hamilton, HH). **d–f** tSNE representations of the three scRNA-seq datasets with "broad" cell type annotations, with the mesenchymal populations used for "fine" re-clustering highlighted in color. **g** Unsupervised hierarchical clustering and heatmap representation of pairwise

Spearman's rank correlation coefficients of highly variable genes in pseudobulk transcriptomes of "broad" cell type clusters. Anatomical origins of pseudobulks are indicated by color code. Mesenchymal cell populations across embryonic origins cluster together (red dotted square). **h–k** "Fine" re-clustering of mesenchymal populations. **h** Dot plot of chondrogenic marker genes (black) and chondrogenic modules (red) expression in "fine" clusters identified across the three embryonic origins. The number of genes contained in the two modules is indicated in brackets. *EC* early chondrogenic, *PC* precursor, *IM* intramembranous ossification, *LS* late skeletal. tSNE representations of re-clustered mesenchymal cells of nasal (**i**), somite (**j**), and limb (**k**) origins, with "fine" cluster annotations and superimposed streamline plots of *scVelo* vector fields. Ec clusters are highlighted in red.

(Fig. 1b). The early specification of these cells thus needs to integrate discrete molecular states, inherited from their respective embryonic PC sources, to facilitate a skeletogenic cell fate convergence.

The distinct embryonic trajectories of these skeletogenic cells also reflect the evolutionary histories of the different parts of the vertebrate skeleton. The first vertebrate skeletal elements are thought to have originated in the head region, mirroring the pre-vertebrate presence of cartilage-like structures supporting a feeding apparatus[19]. Subsequently, the ability to form a progenitor cell-based endoskeleton expanded along the primary and secondary body axes, giving rise to structurally supportive yet flexible elements in the axial and appendicular skeletons, respectively[20]. However, these distinct evolutionary and developmental histories may also challenge our understanding to what extent the convergently specified cells of the vertebrate skeleton can be considered truly homologous[8,21,22].

Here, using single-cell functional genomics along the three distinct mesenchymal PC-to-skeletogenic cell trajectories, we investigate the genome-wide regulatory dynamics in a vertebrate embryo at cellular resolution. We provide evidence that lineage-specific transcription factor profiles are inherited from the respective embryonic origins and that these are integrated at distinct *cis*-regulatory elements to canalize developmental cell fate trajectories toward an early skeletogenic convergence point. These distinct *cis*- and *trans*-dynamics imply a gene regulatory uncoupling between skeletogenic cells at different anatomical locations. We discuss the resulting implications for cell type homology assessment in the vertebrate skeleton and the potential of distinct evolutionary trajectories in skeletal cell and tissue properties upon co-option-dependent convergence of gene regulatory programs across embryonic lineages.

## Results

### A convergent transcriptomic signature in skeletogenic cells of different embryonic origins

To establish the temporal progression of skeletogenic initiation and maturation across the three anatomical locations and embryonic lineages, we first performed chromogenic in situ hybridizations (ISH) on a developmental time series of chicken embryos. We used cranial sagittal cryosections covering the frontonasal prominence and brachial trunk transversal sections, including the emerging forelimb buds. We investigated the expression of *SOX9*, an early marker of skeletogenic induction[15], and *Aggrecan*[23] (*ACAN*), an extracellular matrix protein of more mature skeletal cells (Supplementary Fig. 1a, b). Based on the observed expression dynamics, we devised a sampling strategy following the cell- and tissue-specific transcriptional changes of the three embryonic lineages, from mesenchymal PC toward the onset and maturation of skeletogenic tissues, using *10x Chromium* single-cell RNA-sequencing (scRNA-seq) profiling (Fig. 1c, Supplementary Fig. 1c). Following filtering steps based on different metrics, we obtained a total of over 23,000 high-quality single-cell transcriptomes of comparable complexities (Supplementary Fig. 1d–h). For each of the three anatomical locations, we integrated the three sampled embryonic stages and performed tSNE non-linear dimensionality reduction and graph-based clustering using *Seurat*[24] (Fig. 1d–f). These "broad" clusters were annotated with the help of expression profiles of known marker genes (Supplementary Fig. 2a–c) and showed similar contributions from the three embryonic stages (Supplementary Fig. 2d–f).

To assess transcriptomic similarities amongst these clusters, we next generated cluster-based pseudobulks and calculated Spearman's rank correlation coefficients on differentially expressed genes across all cell types and anatomical locations. Unsupervised hierarchical clustering revealed that cell types originating from the same embryonic lineage, but sampled at different anatomical locations–like, e.g., skin or blood cells–showed highly similar transcriptional profiles, in agreement with their shared developmental history (Fig. 1g). Intriguingly, however, our analysis revealed that also

mesenchymal cells stemming from discrete embryonic lineages–that is, from the NC, the somites, or the LPM–clustered together, indicating a transcriptional convergence amongst them (Fig. 1g, red dotted square). Across our three anatomical sampling sites, we focused on "broad" clusters that likely contained cells transitioning from a mesenchymal PC state toward a skeletogenic fate (Fig. 1d–f, color coded) and re-clustered them at finer resolutions. Within these "fine" clusters, we again used expression profiles of known marker genes, as well as two previously identified early chondrogenic (EC) gene co-expression modules, "IMM"[25] and "RED,"[26] consisting of 17 and 41 genes, respectively, to assess their respective skeletogenic differentiation. Furthermore, we used known markers for the three embryonic PC populations, a general proliferative signature, as well as markers for other skeletogenic cell types, to refine our annotation of these "fine" clusters (Supplementary Fig. 2g–i). In each embryonic lineage, we identified cells enriched for an early skeletogenic, i.e., chondrogenic, signature (Fig. 1h; "EC," highlighted in red), as well as the respective PC populations (Fig. 1h; "PC.") We additionally isolated a cluster showing signs of more mature skeletal cells in the limb sample (Fig. 1h, "LS,") as well as cells with a signature indicative of intramembranous ossification in the nasal sample (Fig. 1h, "IM," Supplementary Fig. 2g). Using *scVelo*[27], we approximated cell fate transition trajectories in silico and projected the predicted vector fields using streamline plots on tSNE representations of our mesenchymal samples. For all three anatomical locations, *scVelo* predicted trajectories connecting the mesenchymal PC–as identified by known marker genes –to EC populations (Fig. 1i–k).

Using ISH and single-cell transcriptomics, our analyses revealed distinct temporal dynamics but converging molecular signatures amongst mesenchymal cells with skeletogenic potential across the three embryonic lineages. Furthermore, our *scVelo* analysis suggested that our single-cell transcriptomics data captured the entire specification spectrum, from uncommitted mesenchymal PC cell to early chondrocyte.

### Distinct *trans*- and *cis*-regulatory modalities underlie the convergent specification of skeletogenic cells

To detail the transcriptional signatures underlying the switch from uncommitted mesenchymal PC cell to early chondrocyte, we focused our analyses on the "fine" mesenchymal clusters with skeletogenic potential. We re-assessed transcriptional similarities of these subpopulations using differentially expressed genes and Spearman's rank correlation coefficients of pseudobulk transcriptomes. Again, we found high transcriptional similarities amongst early skeletogenic cells at the three anatomical locations, as indicated by their clustering together (Fig. 2a, red dotted square).

To investigate potential upstream regulatory inputs controlling these similar transcriptomes, we next focused our attention on transcription factors. Spearman's rank correlation coefficients on transcription factor expression profiles, however, re-clustered the mesenchymal populations strictly by embryonic origins, irrespective of their skeletogenic differentiation state (Fig. 2b). This suggested that the mesenchymal PC populations carried over a lineage-specific repertoire of expressed transcription factors while undergoing skeletogenic induction. Indeed, when looking at transcriptional regulators enriched in chondrogenic cells of the three anatomical locations, we find clear evidence of a lineage-specific heritage of expressed transcription factors, many of which have known developmental functions in their respective anatomical locations (Fig. 2c). Thus, counterintuitively, this indicated that–across anatomical locations and embryonic lineages–overall similar transcriptional signatures were generated with distinct upstream *trans*-regulatory inputs, i.e., lineage-specific transcription factor expression profiles.

We reasoned that these distinct *trans*-regulatory inputs could potentially be integrated at the *cis*-regulatory level of key skeletogenic

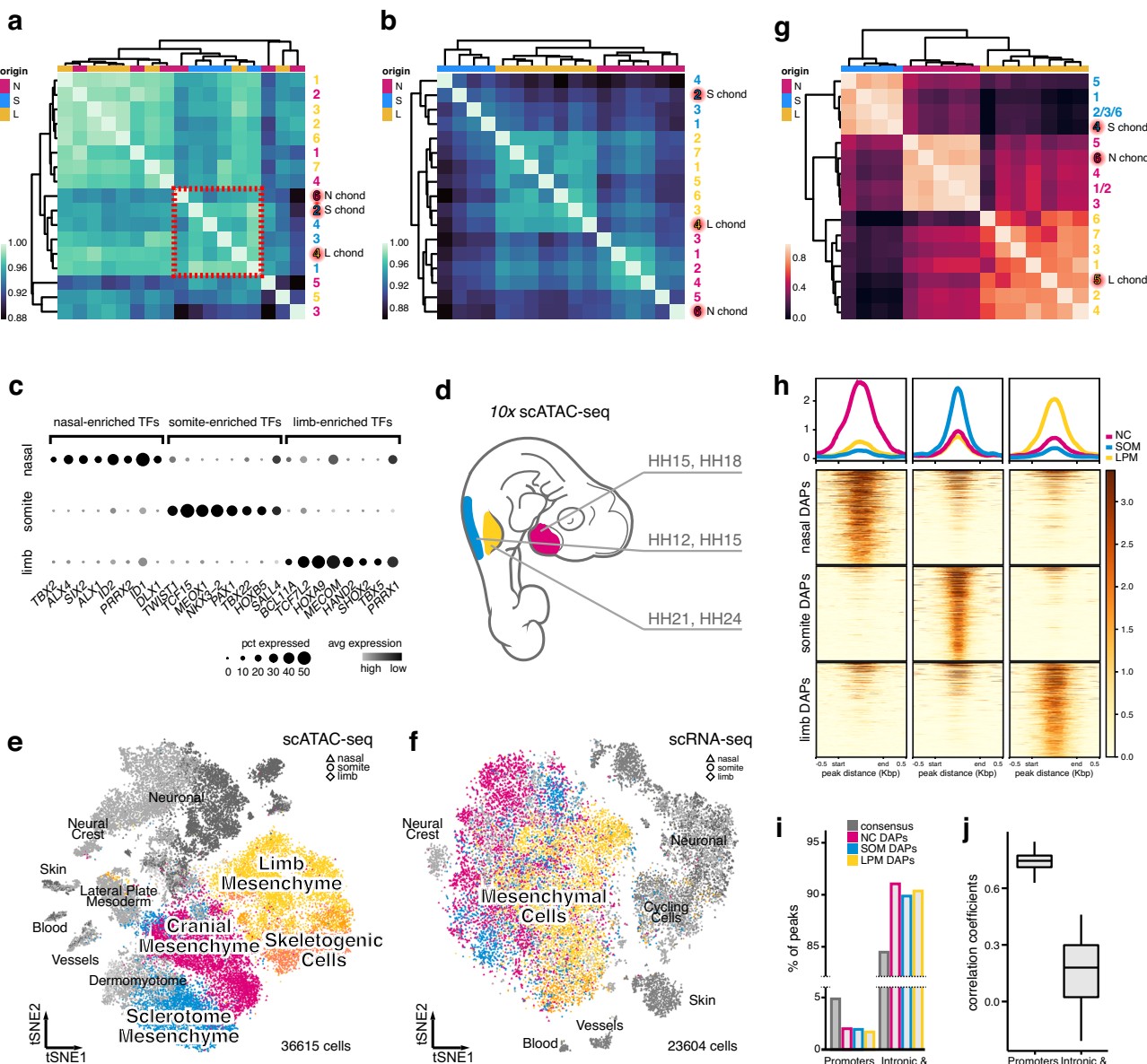

**Fig. 2 | Distinct *trans*- and *cis*-regulatory modalities underlie the convergent specification of skeletogenic cells. a** Unsupervised hierarchical clustering and heatmap representation of pairwise Spearman's rank correlation coefficients of highly variable genes in pseudobulk transcriptomes of "fine" mesenchymal clusters. Anatomical origins of pseudobulks are indicated by color code. EC cells (highlighted in red) across embryonic origins cluster together (red dotted square). **b** Unsupervised hierarchical clustering and heatmap representation of pairwise Spearman's rank correlation coefficients of expressed transcription factors in pseudobulk transcriptomes of "fine" mesenchymal clusters. Anatomical origins of pseudobulks are indicated by color code. All pseudobulks, including EC cells (highlighted in red), cluster by embryonic origins. **c** Dot plot of embryonic origin-enriched transcription factors' expression in EC cells. **d** Sampling scheme to assess chromatin accessibility signatures of skeletogenic cells across anatomical locations (color coded) and embryonic stages (Hamburger–Hamilton, HH). tSNE representations of integrated single-cell chromatin accessibility (**e**) and single-cell transcriptome (**f**) data across the three anatomical locations. The anatomical origins of

cells are indicated by symbols, with the mesenchymal populations used for "fine" re-clustering highlighted in color. **g** Unsupervised hierarchical clustering and heatmap representation of pairwise Spearman's rank correlation coefficients of differentially accessible peaks in pseudobulk chromatin accessibility data of "fine" mesenchymal clusters. Anatomical origins of pseudobulks are indicated by color code. All pseudobulks, including EC cells (highlighted in red), cluster by embryonic origins. **h** Coverage plots and heatmap representations of the top 500 differentially accessible peaks (DAPs) in EC cells of neural crest, somatic, and LPM origin. **i** Promoter depletion and intronic/intergenic elements enrichment of DAPs in EC cells, relative to the consensus peak set. **j** Boxplots of pairwise Spearman's rank correlation coefficients of DAPs in EC cells across embryonic origins, calculated using promotor–proximal elements ($n = 1023$) or an equal number of randomly sampled distal intronic/intergenic elements. Boxplots of 83 pairwise Spearman's rank correlation coefficients, center line = median, box limits = quartiles, and whiskers = 1.5× interquartile range.

genes to facilitate transcriptomic convergence. To investigate this possibility, we performed single-cell chromatin accessibility assays (single-cell assay for transposase-accessible chromatin with sequencing, or scATAC-seq[28]) to identify genomic elements with potential regulatory activity. We followed a similar sampling scheme as for our scRNA-seq approach, although—reasoning that such *cis*-regulatory

recoding would occur during early stages of skeletogenic induction—we excluded the late time points (Fig. 2d). In total, we obtained over 36,000 cells with high-quality chromatin accessibility profiles and, using *MACS2*[29], we identified a consensus set of 678,707 peaks across the three anatomical locations (Supplementary Fig. 3a–c). We integrated the two embryonic stages per sampling site, performed non-

linear dimensionality reduction, and identified potential cell type-specific clusters (Supplementary Fig. 3d–f). We annotated these clusters with the help of our scRNA-seq data, using label transferring and non-negative least squares (NNLS) regression, as well as visual inspection of scATAC-seq "marker peaks" (see Methods and Supplementary Fig. 3g). At all three anatomical locations, overall similar cell type repertoires were recovered as in our scRNA-seq sampling.

We then combined all samples across the three anatomical locations for both scRNA-seq and scATAC-seq data sets and performed anchor-based integration. Mesenchymal cells in our scATAC-seq data visually appeared to intermingle less than in our scRNA-seq data (Fig. 2e, f). This may indicate the presence of embryonic origin-specific states in chromatin accessibility, as opposed to the convergent signatures at the transcriptomic level (Fig. 1g). To investigate potential lineage-specific chromatin accessibilities, we re-clustered the mesenchymal scATAC-seq cell populations with skeletogenic potential and annotated the resulting "fine" clusters using our scRNA-seq data (Supplementary Fig. 3h–j). We then performed unsupervised hierarchical clustering on Spearman's rank correlation coefficients of differentially accessible peaks (DAPs) in pseudobulks of these "fine" mesenchymal clusters. Akin to our scRNA-seq analyses of transcription factor expression profiles, this resulted in a strict embryonic origin-dependent clustering of mesenchymal populations, including EC cells (Fig. 2b, g). Major cell type classifications, however, still clustered according to their "broad" annotations, with similar chromatin accessibility signatures echoing their shared embryonic origins (Supplementary Fig. 3k). This further implied that mesenchymal and skeletogenic cells across the three anatomical locations carry distinct chromatin accessibility profiles, reflecting their discrete embryonic origins and lineage histories.

Indeed, coverage plots across the three anatomical locations revealed that a substantial fraction of DAPs within the respective chondrogenic populations are distinct and, hence, embryonic origin-specific (Fig. 2h). We classified these DAPs according to their genomic location concerning the transcription start sites of neighboring genes. Interestingly, we found that promoter-proximal peaks were depleted in our DAP set, relative to the consensus gene set, while more distal elements—intronic and intergenic—appeared enriched (Fig. 2i). This implied that most differences in chromatin accessibilities, between chondrogenic cells from different embryonic lineages, are found at distally located peaks. Indeed, DAPs at promoter-proximal peaks showed, on average, higher Spearman's rank correlation coefficients across embryonic origins than distal intronic/intergenic ones (Fig. 2j). This suggested that the similar transcriptional profiles observed at the RNA level originate from similar promoter repertoires. Distal peaks, however, where one would anticipate putative long-range enhancer elements to be located, showed much lower similarity in accessibilities amongst the different embryonic origins (Fig. 2j).

Collectively, our combined scRNA-seq and scATAC-seq approach revealed the presence of distinct *trans-* and *cis-*regulatory signatures in skeletogenic cells at the three anatomical locations, as evidenced by embryonic origin-specific transcription factor expression profiles and the presence of discrete chromatin accessibility signatures at distal locations.

### *Trans-* and *cis-*regulatory dynamics of skeletogenic convergence across the three embryonic lineages

To follow the *trans-* and *cis-*regulatory changes underlying these convergent cell fate transitions, we next investigated scRNA-seq and scATAC-seq dynamics along chondrogenic pseudotime trajectories in the three anatomical locations. Using tSNE embeddings of our scRNA-seq data, we constructed minimum spanning trees using *slingshot*[30], with preset start points corresponding to the clusters with naïve mesenchymal expression signatures (Fig.3a–c). We observed overall similar trajectory predictions as for our *scVelo* analysis (Fig.1h–j) and

confirmed their overall topography using another orthogonal approach, *scFates*[31], projected on *ForceAtlas2*[32] graph layouts, for improved visual resolution of the transcriptionally similar clusters (Supplementary Fig. 4a–c). Overall, for each anatomical location, we were able to retrieve a single trajectory either ending in an EC cluster (Fig. 3a, b), or traversing it toward more mature skeletal cells (Fig. 3c). We then transferred the pseudotime values of our chondrogenic scRNA-seq trajectories to our scATAC-seq data, to follow the accompanying chromatin accessibility dynamics (Fig. 3a–c, insets).

We binned the respective scRNA-seq pseudotimes into equidistant pseudobulks and used *TrAGEDy*[33] to align the chondrogenic gene expression dynamics along the respective trajectories of the three embryonic origins. We found overall higher similarities toward the ends of these pairwise comparisons, indicating chondrogenic convergence at the transcriptional level (Supplementary Fig. 4d–f). Both nasal and somite trajectories failed to align to the last section of our limb chondrogenic pseudotime. This corroborated the notion that we had recovered more mature skeletal cells in our limb samples compared to nasal and somite (Supplementary Fig. 4g–i, Fig. 1h). Accordingly, we excluded the corresponding limb pseudotime bins containing these cells from further analyses.

We first checked the expression dynamics of the two chondrogenic gene co-expression modules "IMM" and "RED." Both modules increased in their expression along the three pseudotime trajectories in nasal, somite, and limb samples (Fig. 3d–f, red). Next, we looked at the expression dynamics of a core set of common chondrogenic genes, which were found to be enriched in chondrocytes across the three embryonic lineages (Supplementary Fig. 5a). All genes showed an increase in expression along the respective trajectories (Fig. 3d–f, black). Contained within this shared set of genes were known chondrogenic regulators and extracellular matrix proteins, as well as a selection of ribosomal proteins (Fig. 3d–f, Supplementary Fig. 5a). Additionally, using *tradeSeq*[34], we identified lineage-specific expression dynamics of shared and distinct regulators activated in chondrogenesis across the three anatomical locations (Supplementary Fig. 5b–d). In all three lineages, the chondrogenic wave appeared to be preceded by an increase in expression of origin-specific transcription factors (Figs. 2c and 3g–i). Finally, we found evidence for distinct chromatin accessibility dynamics. Chondrocyte-specific DAPs showed increased accessibility along the respective chondrogenic scATAC pseudotime trajectories but in an embryonic origin-specific manner (Fig. 3j–l).

Using integrative scRNA-seq and scATAC-seq pseudotime analyses, we detailed the emergence of common transcriptional signatures and distinct *trans-* and *cis-*regulatory profiles during skeletogenic convergence. Namely, while increased expression of chondrogenic modules and a core set of differentially expressed genes was shared across the three embryonic origins, these were accompanied by lineage-specific transcription factor expression and chromatin accessibility dynamics.

### Embryonic origin-specific activities and specificities of transcription factor binding motifs and transcription factor-protein interaction profiles

To investigate the interplay of distinct transcription factor profiles and origin-specific chromatin accessibilities, we next evaluated cell type-specific activities of transcription factor binding motifs. Given the scarcity of publicly available and experimentally validated binding motifs for chicken transcription factors, we decided to define our own set of DNA position weight matrices. Briefly, we used *Homer*[35] to identify enriched de novo motifs in our scATAC-seq data in a cluster-by-cluster manner across the three anatomically distinct samples. We annotated these de novo motifs with candidate transcription factors using public repositories and selected the best matches based on motif similarity and the correlation of motif

 

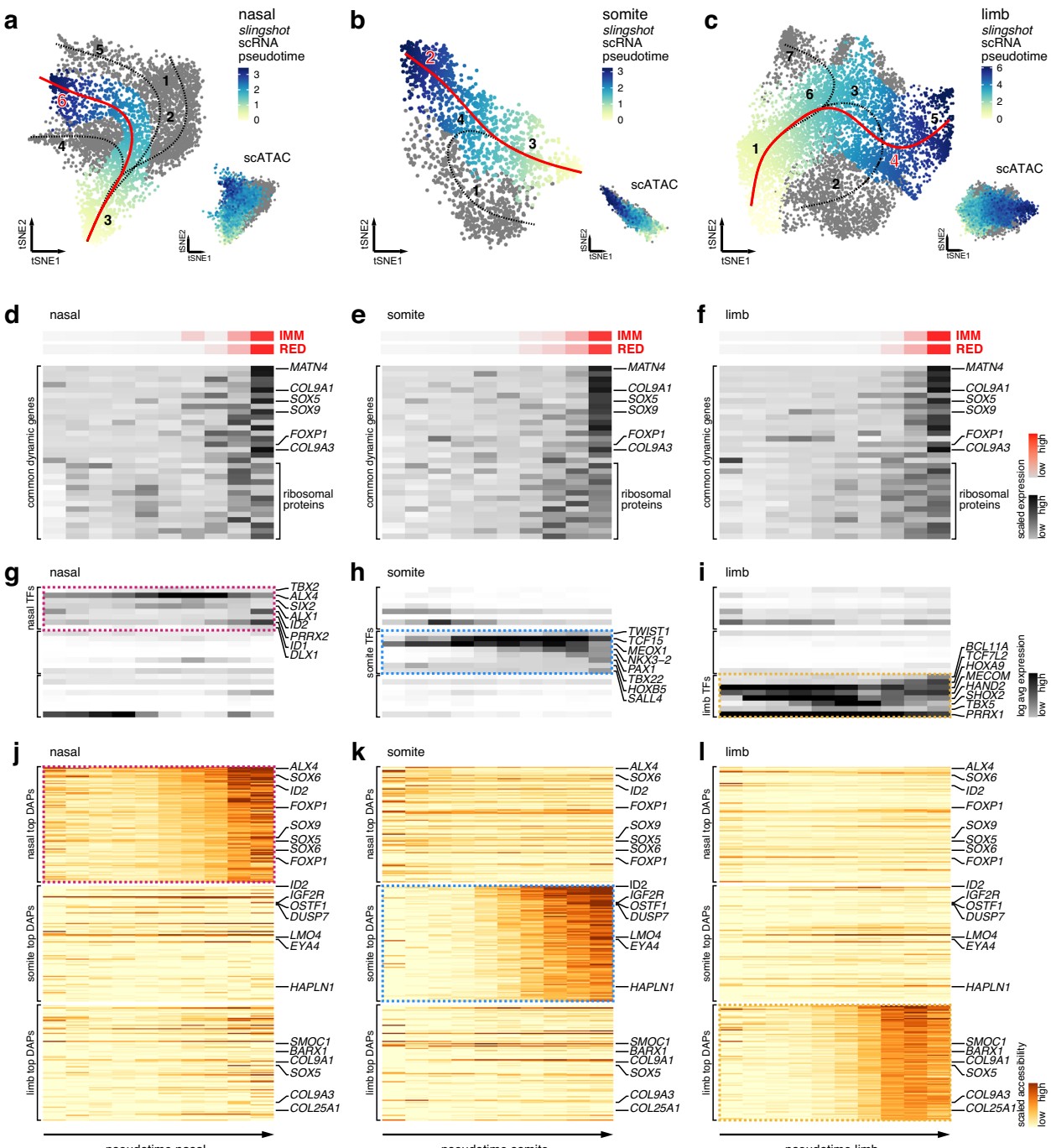

**Fig. 3 | *Trans*- and *cis*-regulatory dynamics of skeletogenic convergence across three embryonic lineages.** tSNE representations of single-cell transcriptomes and co-embedded single-cell chromatin accessibilities (insets) for nasal (**a**), somite (**b**), and limb (**c**) origins. Superimposed on the single-cell transcriptomes are the pseudotime trajectories identified by *slingshot*, with the chondrogenic trajectories used for further analyses highlighted in red. Pseudotime progression is visualized by heatmaps on scRNA and scATAC data. "Fine" cluster annotations as in Fig. 1, with EC clusters highlighted in red. **d–l** Binned pseudobulk dynamics along chondrogenic pseudotime trajectories. **d–f** Z-score scaled expression dynamics along the chondrogenic pseudotime trajectories for chondrogenic modules (red) and common differentially expressed genes identified in all three embryonic origins (black). **g–i** Log-transformed expression dynamics of embryonic origin-specific transcription factors identified in nasal (magenta dotted line), somite (blue dotted line), and limb (yellow dotted line) samples. **j–l** Average normalized peak accessibility dynamics of embryonic origin-specific differentially accessible peaks (DAPs) identified in nasal (magenta dotted line), somite (blue dotted line), and limb (yellow dotted line) samples. Select DAP-adjacent genes are indicated on the right.

activities with scRNA-seq transcription factors' expression profiles (see Methods). In total, we identified and annotated 1373 de novo motifs across the three anatomical locations (Supplementary Fig. 6a), with 540 non-redundant ones used for the further analyses (Fig. 4a). Motifs for members of the homeobox, C2H2 zinc fingers and basic helix-loop-helix (bHLH) protein transcription factor families were amongst the most frequently identified ones (Fig. 4a). Furthermore, our limb mesenchyme de novo motif for SOX9 matched a chick limb ChIP-seq[36] validated motif more closely than publicly available position weight matrices (Supplementary Fig. 6b). Encouraged by this, we continued all subsequent analyses with our de novo motifs only.

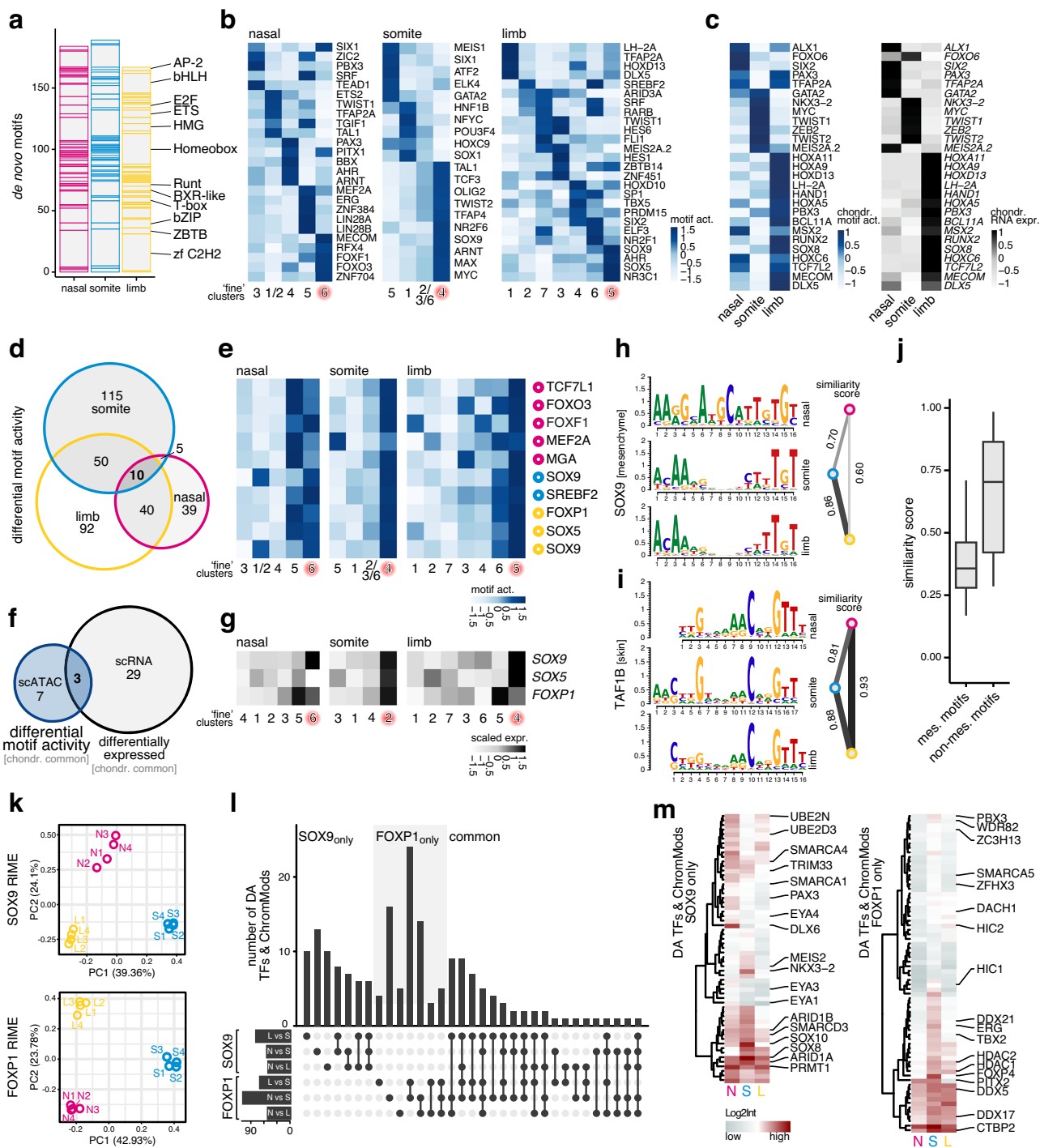

Using this custom set of position weight matrices, we next conducted differential motif activity analyses. At both "broad" and "fine" cluster resolutions, we identified motif activity signatures that were predictive for specific cell types, including early chondrocytes and mesenchymal PC cells (Fig. 4b and Supplementary Fig. 6c, d). Moreover, we identified motif activities in chondrogenic cells with high embryonic origin-specificity. These were mirrored by expression signatures of the corresponding transcription factors. This emphasized the presence of distinct *trans*-regulatory inputs in chondrogenic cells of the three embryonic lineages, at both RNA and motif activity levels (Fig. 4c). Of all motifs whose activities were enriched in chondrogenic cells, only ten were shared across the three anatomical locations (Fig. 4d, e, Supplementary Fig. 6e). Comparing them to the core chondrogenic genes identified in our differential expression analyses

revealed only three genes showing consistent chondrocyte enrichment at both RNA expression and motif activity levels: *SOX9, SOX5,* and *FOXP1* (Fig. 4f, g).

With the length range of our *Homer* de novo motif search (8–22 bp), we were able to investigate the occurrence of potential co-binding patterns of multiple transcription factors. Sequences bound by multimeric protein complexes are increasingly recognized as an integral aspect of DNA's regulatory grammar to provide robustness or diversify activity patterns in a combinatorial manner[37–39]. Indeed, many of our motifs showed a bimodal distribution of nucleotide enrichment in their position weight matrices, indicative of dimers binding to them. For example, the SOX9 motifs identified in somite and limb mesenchymal cells indicated a homodimer-like binding, in agreement with previous findings[36] (Fig. 4h and Supplementary Fig. 6b).

**Fig. 4 | Embryonic origin-specific transcription factor binding motif activities and protein interaction profiles. a** Final number and transcription factor (TF) family distribution of non-redundant de novo identified binding motifs across the three embryonic origins. **b** Differential cluster-specific motif activities in mesenchymal populations across embryonic origins. EC clusters are highlighted in red. **c** Embryonic origin-specific TF motif activities (left) and mRNA expression profiles (right), enriched in EC cells. **d** Venn diagram displaying the overlap of chondrocyte-enriched motif activities across embryonic origins. **e** Motif activity heatmaps of the ten commonly chondrocyte-enriched TF motifs across embryonic origins. The embryonic origin in which the respective motif was identified is indicated by color-coded circles on the right. EC clusters are highlighted in red. **f** Venn diagram displaying the overlap of unique chondrocyte-enriched motif activities and commonly enriched chondrogenic genes. **g** Enriched expression of *SOX9*, *SOX5*, and *FOXP1* in EC cells of the three embryonic origins. Position weight matrices of a TF motif identified in mesenchymal cells of all embryonic origins (SOX9, **h**) and a TF motif identified in non-mesenchymal cells (skin) of all

embryonic origins (TAF1B, **i**). Pairwise motif similarity scores are displayed on the right, with wider/darker lines indicating higher similarity. **j** Boxplots of pairwise motif similarity scores for TFs identified in multiple embryonic origins. Motifs were binned according to which general cell population they were identified in, i.e., mesenchymal ($n = 204$) *versus* non-mesenchymal ($n = 19$). Boxplot center line = median, box limits = quartiles, and whiskers = 1.5× inter-quartile range. **k** Principal component analysis of replicate SOX9 and FOXP1 RIME (Rapid immunoprecipitation mass spectrometry of endogenous proteins) experiments in NC- (N, magenta), somitic mesoderm- (S, blue), and LPM-derived (L, yellow) tissues. Numbers in brackets indicate the percentage of total variance explained by PC1 and PC2. **l** *UpSet* plot of significantly differentially abundant (DA) transcription factors and chromatin modifiers identified pairwise across pairs of embryonic origins for SOX9 and FOXP1 RIME experiments. **m** Averaged log2-normalized intensities for SOX9-specific (left) and FOXP1-specific (right) DA transcription factors and chromatin modifiers across embryonic origins.

---

Additionally, in nasal samples, our analyses predicted a motif showing a SOX9-like monomer signature at its 3′ end, but a potential heterodimeric binding partner at its 5′ end (Fig. 4h). To investigate this phenomenon more systematically—that is, the same transcription factor having dissimilar binding motifs predicted in the three different embryonic lineages—we trimmed the extremities of our position weight matrices based on minimal nucleotide enrichment scores and calculated motifs similarities across anatomical locations. In our analysis, we split the motifs based on whether their position weight matrices were identified in mesenchymal cells of different embryonic origins, or non-mesenchymal populations of the same embryonic lineage (e.g., skin, Fig. 4i). On average, position weight matrices for transcription factors identified in mesenchymal cells showed lower similarity scores than motifs identified in non-mesenchymal cell types (Fig. 4j). This indicated that binding motifs for the same transcription factors were less conserved in mesenchymal cells of different embryonic origins, potentially due to lineage-specific differences in their cells' chromatin environment, or distinct co-factors binding the motifs in a heterodimeric manner (Fig. 4h–j).

To follow up on these observations, we focused our attention on two of the core factors we identified as chondrocyte-enriched at both motif activity and RNA expression levels, SOX9 and FOXP1 (Fig. 4f, g), and performed Rapid immunoprecipitation mass spectrometry of endogenous proteins[40] (RIME). RIME allows for mass spectrometry-based identification of protein assemblies and thus offers an experimental approach to probe for potential lineage-specific differences in the composition of transcription factor complexes (see Methods and Supplementary Data 1). We assessed the specificity of our two antibodies against SOX9 and FOXP1 in a limb tissue test run (Supplementary Fig. 7a–d), and identified previously reported protein interactors, such as e.g., CTNNB1[41] or members of the SWI/SNF complex (e.g., ARID1A/B and SMARCD1/2)[42] (Supplementary Fig. 7d and Supplementary Data 1). We then performed quadruplicate RIME experiments for SOX9 and FOXP1, probing the skeletogenic cells of all three embryonic lineages. Overall, we detected 1475 (SOX9) and 960 (FOXP1) proteins, many of which were shared between the two factors (Supplementary Fig. 7e–g). Within each experiment, however, the composition of the co-bound proteins revealed clear signals of their embryonic origins (Fig. 4k). To identify potential lineage-specific interactors, we first performed DA analyses across the three embryonic origins and focused on DA proteins specific to either SOX9 or FOXP1 pairwise comparisons[43] (Supplementary Fig. 7h–n and Supplementary Data 1). We further subsetted this list for transcription factors and chromatin modifiers, reasoning that interactors from these protein classes were most likely to modulate the DNA binding patterns of our two candidates (Fig. 4l, Supplementary Fig. 7o). For both SOX9 and FOXP1, we identified proteins that were enriched in our pulldowns across the embryonic origins, albeit to different degrees. These were

mostly general transcriptional regulators involved in chromatin remodeling or co-transcriptional RNA processing (Fig. 4m, Supplementary Fig. 7p). Intriguingly, however, we also identified several proteins that appeared to specifically interact with one of our candidates only and in a lineage-specific manner. Examples include the EYA proteins in neural crest-derived tissues as well as MEIS2 and NKX3-2 in somitic tissues for SOX9, or the paralogs HIC1 and HIC2 in our FOXP1 pulldowns from somites and limbs, respectively (Fig. 4m, Supplementary Fig. 7p).

Collectively, we identified de novo transcription factor binding motifs, some of which showed common cell type-specific activities while others were embryonic lineage-restricted. We find partially diverging sequence logos for the same transcription factors in mesenchymal cells at different anatomical locations. Furthermore, using immunoprecipitation and proteomics-based identification of co-bound proteins for two candidates, SOX9 and FOXP1, our experiments suggest that chondrogenic transcription factors can interact with distinct co-regulators in a lineage-specific manner. This substantiates our previous findings of lineage-specific *trans*-regulatory inputs during the skeletogenic convergence of different embryonic PC lineages at the level of motif architectures and activities, as well as putative binding partners in transcription factor complexes.

## Lineage-specific regulatory architectures in skeletogenic cells and *cis*- and *trans*-regulatory dynamics during chondrogenic induction

We used *ArchR*[44] on our mesenchymal scATAC-seq data for peak-to-gene links analyses to connect putative distal enhancer elements to target genes and test for lineage-specific activities. Based on a minimal correlation coefficient and adjusted p-value cutoffs, we identified over 28,000 presumptive peak-to-gene links (Fig. 5a–c and Supplementary Fig. 8a–f). Within these peak-to-gene links, target genes were generally predicted to be contacted by less than three putative *cis*-regulatory elements (CRE), and the majority of CREs interacted with only one target gene (Supplementary Fig. 8d–f). Using hierarchical k-means (hkmeans) clustering, we sorted our peak-to-gene links according to their aggregate activity profiles and plotted z-score normalized chromatin accessibilities and imputed target gene expression levels (Fig. 5a–c and Supplementary Fig. 8g–i). For both somite and limb samples, and to some extent in nasal cells, we identified clusters enriched for EC cells (Fig. 5a–c), with corresponding enrichments for skeletogenesis-related terms (Supplementary Fig. 8g–i).

Of all peak-to-gene links, only 40 CREs were detected in all three anatomical locations using our stringent cutoffs. However, that number increased to 607, if we were only focusing on the overlap of predicted target genes therein (Fig. 5d). Among our peak-to-gene links, the common CREs that were shared between all embryonic origins were on average more conserved across vertebrates, hinting that

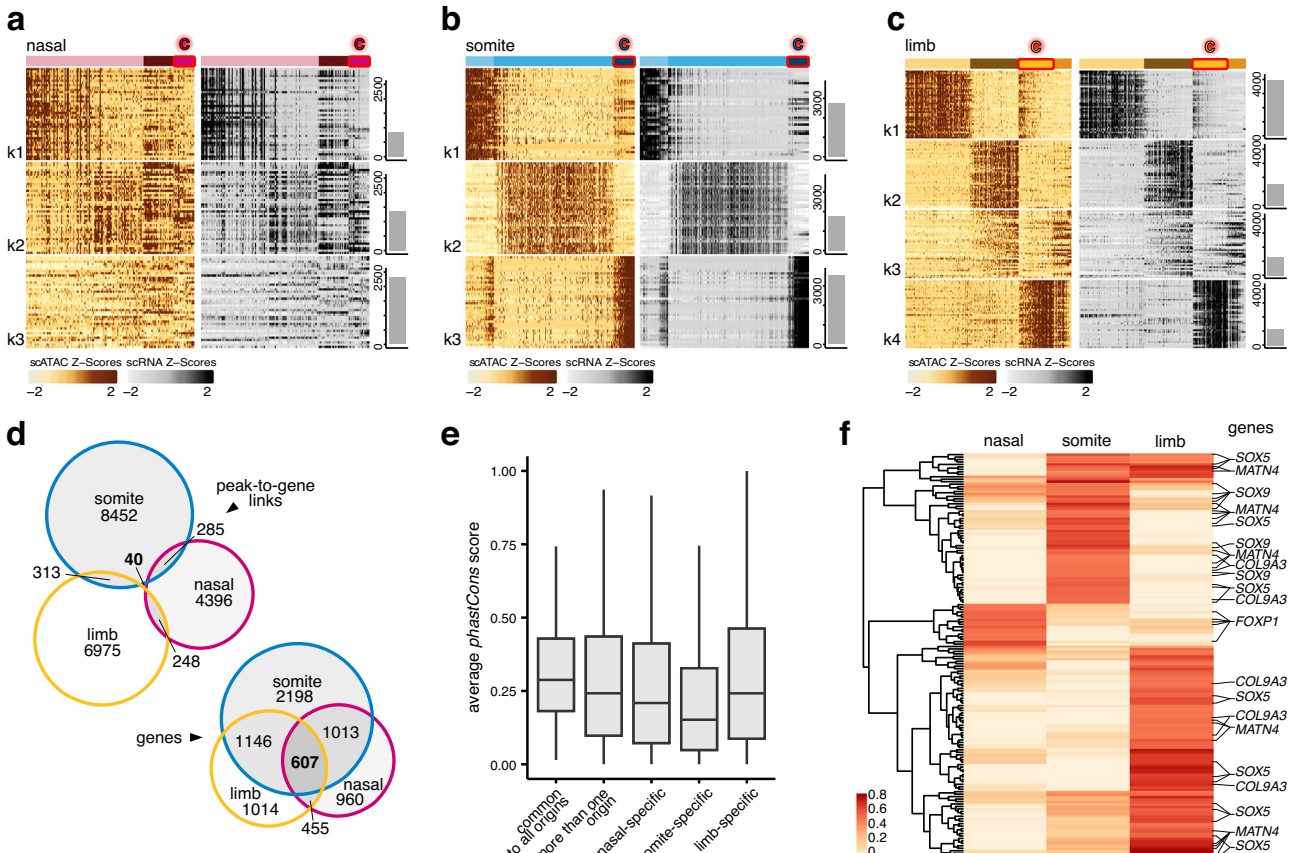

**Fig. 5 | Lineage-specific enhancer-promoter interactions of core chondrogenic genes. a-c** hkmeans-clustered peak-to-gene link heatmaps displaying Z-score normalized peak accessibilities (left, ocher) and imputed target gene expression levels (right, gray) in single cells (columns). Cell clusters dominated by EC cells are highlighted in red (top). The top 40 peak-to-gene links are shown. Total numbers of identified peak-to-gene links per cluster are indicated by barplots on the right, cluster numbers (kX) on the left. **d** Venn diagram displaying the overlap of mesenchymal peak-to-gene link CREs across embryonic origins (top) and the

overlap of target genes contained within these peak-to-gene links (bottom). **e** Average evolutionary conservation (calculated as *phastCons* scores) across CREs in peak-to-gene links shared amongst all embryonic origins ($n = 40$), shared by more than one embryonic origin ($n = 886$), and that are origin-specific ones (nasal $n = 4396$, somite $n = 8452$, and limb $n = 6975$). Boxplot center line = median, box limits = quartiles, and whiskers = 1.5× inter-quartile range. **f** Hierarchically clustered peak-to-gene link correlation heatmap of links identified at core chondrogenic genes across embryonic origins. Select target genes are indicated on the right.

pleiotropic constraints might restrict their sequence substitution rates through purifying selection (Fig. 5e). Furthermore, on average ~15% of our CREs contained an "avian-specific highly conserved element" (ASHCEs[45], nasal 12.9%, somite 15.9%, limb 15.0%, and common 7.5%), but only few "chicken accelerated regions" (CARs[46], 0.09% overall (19 out of 20,709 CREs total)). This suggested that many core genes commonly expressed in vertebrate chondrogenic cells were contacted by embryonic lineage-specific enhancer elements, a subset of which appeared to contain sequences that have become and remained highly conserved specifically during avian evolution. Indeed, looking at the correlation scores of our core chondrogenic gene set (Supplementary Fig. 5a, excluding ribosomal proteins) revealed largely lineage-specific peak-to-gene link activities (Fig. 5f). We further explored the resulting lineage-specificity in *cis*- and *trans*-regulatory dynamics at the *SOX9* locus, a conserved regulator of vertebrate chondrogenesis[15].

While a high density of consensus peaks was present in the genomic landscape flanking *SOX9*, the overall pseudobulk chromatin accessibility profiles looked distinct from one anatomical location to another, both in mesenchymal PC (bright colors) as well as at the EC stage (dark colors) (Fig. 6a). Using HOMER and our custom set of position weight matrices, we identified transcription factors whose binding motifs were enriched in lineage-specific DAPs situated within a 1 Mb (megabase) interval around the SOX9 locus (Fig. 6b). We then followed their genome-wide motif activities and RNA expression

profiles along our scATAC-seq and scRNA-seq pseudotimes, within the respective embryonic lineages. Both modalities displayed largely congruent temporal activation patterns (Fig. 6c, d). Furthermore, these dynamics suggested distinct temporal hierarchies for the putative *trans*-regulatory inputs orchestrating the lineage-specific activation of SOX9. At the *cis*-regulatory level, link plots at the locus revealed that many of the peak-to-gene links were predicted to be embryonic origin-specific (Fig. 6e). To evaluate putative enhancer functions of these peaks and test for their lineage-specificity, we first used differential chromatin accessibility analysis to define chondrocyte-enriched DAPs at the *SOX9* locus. For all three anatomical locations, we identified peaks with high chromatin accessibilities in EC cells, relative to mesenchymal PC and chondrocytes of other embryonic origins (Fig. 6f). We isolated the corresponding sequences from genomic DNA and cloned them into reporter plasmids, upstream of a minimal promoter driving green fluorescent protein (GFP) expression. We electroporated (EP) the resulting constructs into the PC populations of the respective skeletogenic lineages *in ovo*. As EP control, we included a plasmid containing a strong constitutively active promoter driving tdTomato expression. Post-EP, we let the embryos develop further for two days, harvested the targeted tissues, and processed them for histology. We performed immunohistochemistry against endogenous SOX9 protein to determine the location of chondrogenic condensations and against

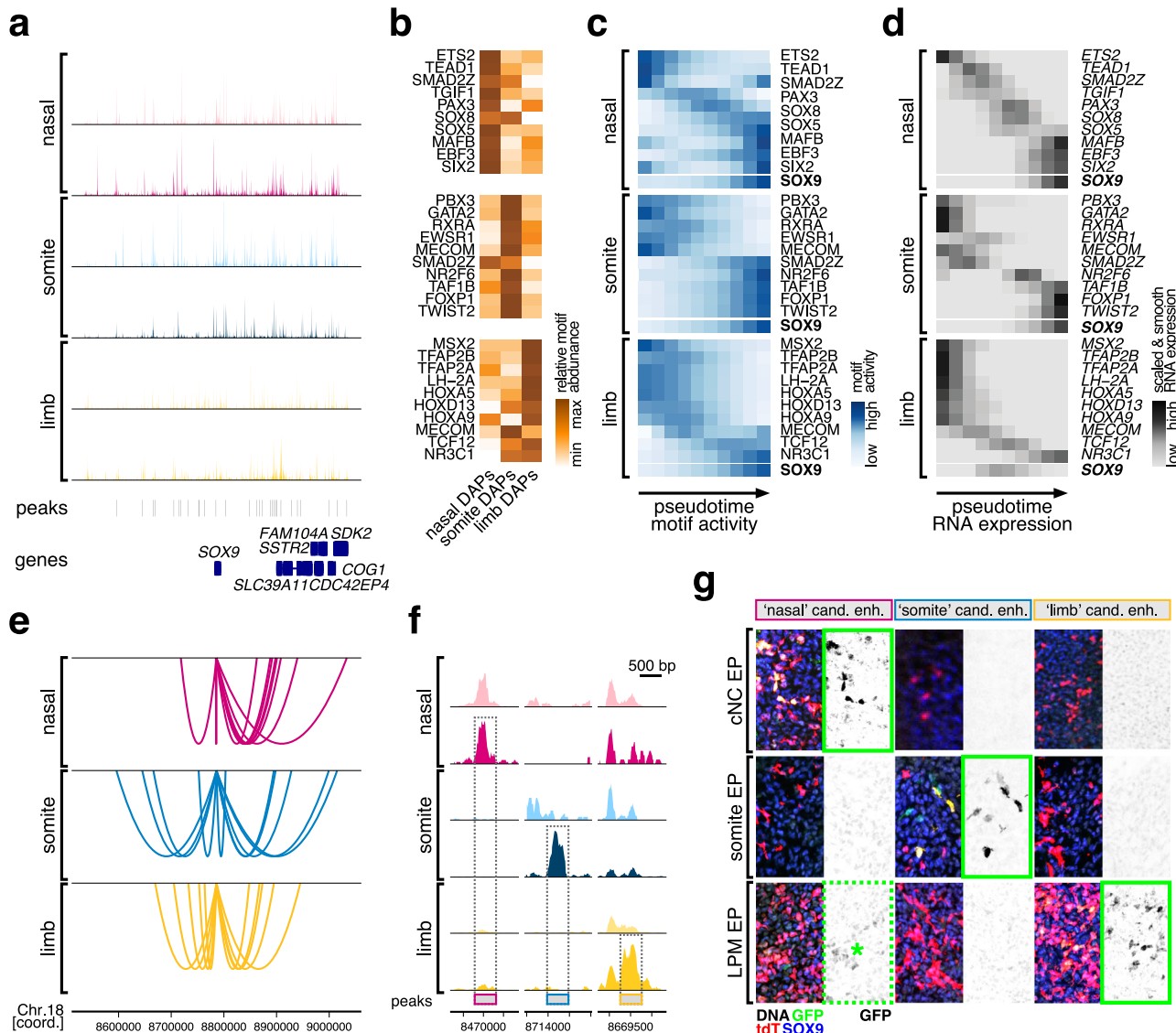

**Fig. 6 | Distinct *cis-* and *trans*-regulatory dynamics during the lineage-specific activation of the core chondrogenic transcription factor *SOX9*. a** Pseudobulk aggregate genome accessibility tracks at the *SOX9* locus. For each embryonic origin, tracks for PC populations (light colors) and EC cells (dark colors) are displayed. Identified peaks and neighboring genes are indicated below. **b** Relative transcription factor binding motif abundances in DAPs at the *SOX9* locus in cells of nasal (left), somite (middle), and limb origin (right). Genome-wide dynamics of motif activities (**c**) and the corresponding RNA expression profiles (**d**) of transcription factors identified in (**b**), following chondrogenic pseudotime trajectories in nasal (top), somite (middle), and limb cells (bottom). For reference, both motif activities and RNA expression profiles of *SOX9* are included. **e** embryonic origin-specific

peak-to-gene link plots at the *SOX9* locus. **f** Pseudobulk aggregate genome accessibility tracks of peaks tested in enhancer reporter assays in (**g**). Candidate enhancers are highlighted by gray dotted boxes. PC population tracks in light colors, EC tracks in dark colors. **g** In vivo enhancer reporter assays. Chondrocyte candidate enhancer elements with predicted embryonic origin-specificity were cloned into reporter constructs driving GFP expression and electroporated (EP) together with a tdTomato control plasmid into cells of the cranical neural crest (cNC), the somites, or the forelimb LPM. Embryonic origin-specificity of GFP expression in SOX9-positive chondrogenic condensations is indicated by green squares. The candidate "nasal" enhancer showed a weak GFP signal in the limb mesenchyme as well (green dotted square, asterisk).

tdTomato and GFP to evaluate EP efficiency and enhancer activity and specificity, respectively. Within EP-positive condensations, we scored the presence or absence of GFP signal to indicate chondrogenic enhancer activity (Supplementary Fig. 9a–f). The predicted somite and limb enhancers showed specific GFP reporter activity, restricted to the respective embryonic lineages they were identified in (Fig. 6h, green boxes). The nasal candidate enhancer reporter drove a strong GFP signal in cranial neural crest-derived chondrogenic tissue (Fig. 6h, green box). It additionally did so at reduced levels in limb condensations as well (Fig. 6h, green dotted box and asterisk), suggesting only partial lineage-specificity and a slightly more promiscuous enhancer activity for this element[47,48].

Overall, our peak-to-gene link analyses uncovered the presence of lineage-specific CREs contacting an overlapping set of target genes across the three anatomical locations. Furthermore, pseudotemproal motif activities and transcription factor expression profiles, as well as enhancer reporter assays at the *SOX9* locus, suggest a partial lineage dependency in the *cis-* and *trans*-regulatory dynamics driving the activation of target genes in EC cells of different embryonic origins.

## Discussion

Cell type specification in animals relies on the execution of distinct gene regulatory programs during embryonic and post-embryonic development. Consequently, cell type evolution depends on the

origination of new regulatory modalities, to drive innovations in cellular form and function. Here, we have documented the gene regulatory dynamics underlying the embryonic specification of skeletogenic cells in vertebrates, an iconic cell type central to their evolutionary success. Following their first evolutionary appearance at the base of vertebrates, additional embryonic lineages have subsequently acquired the ability to form skeletal cell types. This makes the underlying gene regulatory programs interesting from both evolutionary and developmental perspectives. Namely, how the three embryonic lineages have acquired skeletogenic competency at a gene regulatory level during vertebrate evolution, and, developmentally, how these lineages are transcriptionally recoded during the earliest steps of skeletogenesis, to converge from distinct molecular profiles of their respective origin populations toward similar skeletogenic phenotype (Fig. 1a).

## Transcriptional recoding and lineage-memory in mesenchymal cells

Using scRNA-seq profiling, we have followed the early transcriptional dynamics of cell fate specification in the vertebrate skeleton across its three distinct embryonic lineages (Fig. 1b). We find shared transcriptional signatures between both mesenchymal and skeletal cells of the three embryonic origins (Fig. 1b, g). Transitioning toward such mesenchymal-like signatures may transcriptionally prime different PC lineages to increase their cell fate plasticity, akin to what is observed in various types of metastatic cancers, while still maintaining partially distinct transcriptional and chromatin memories of their embryonic origins[49–51]. Indeed, all three skeletogenic lineages undergo an epithelial-to-mesenchymal transition (EMT), before migrating to the periphery where they condense and appear to initiate a core skeletogenic program, which—at a global scale—shows high similarity in its overall gene regulatory state[25,52,53] (Fig. 1h). We also noted a shared increase in ribosomal protein transcription, potentially to meet the increased translational needs for extracellular matrix protein production and secretion[54] (Supplementary Fig. 5a). Importantly, however, our cellularly resolved trajectories also uncovered embryonic lineage-specific *trans*- and *cis*-regulatory dynamics underlying the early cell fate specification in the vertebrate skeleton. As such, our study closes a crucial gap between skeletal pre-patterning[46] and tissue maturation[25,52,53] across all three embryonic lineages and at cell lineage resolution.

## Distinct *trans*- and *cis*-regulatory dynamics along skeletogenic differentiation trajectories of different embryonic origins

We find a lineage-specific heritage in transcription factor expression profiles in skeletogenic cells of different anatomical locations (Fig. 2b, c). This implies that the convergent specification of functionally analogous and transcriptionally similar skeletal cell types can be induced by distinct upstream *trans*-regulatory inputs, even across germ layers, reminiscent of the functional convergence identified in the developing visual system of *Drosophila*[55]. Here, using scATAC-seq data, we add an additional layer of regulatory information, and demonstrate that distinct chromatin accessibility signatures accompany these specific *trans*-regulatory inputs (Fig. 2h, h), with promoter accessibilities exhibiting overall higher similarities than distal sites with putative enhancer functions (Fig. 2i, j). We followed the *trans*- and *cis*-regulatory dynamics along pseudotemporal skeletogenic trajectories of the three embryonic origins and identified distinct signatures underlying this transcriptional and phenotypic cell fate convergence (Fig. 3d–i, Supplementary Fig. 5a–d). Intriguingly, the expression levels of lineage-specific *trans* regulators (Fig. 3d–i), as well as corresponding binding motif activities (Fig. 6b–d), peak right before the onset of the core skeletogenic program. This suggests the presence of lineage-specific transition states, in which mesenchymal PC are transcriptionally recoded toward a similar skeletogenic cell fate[56]. Thereafter, known skeletal regulators and effector genes become

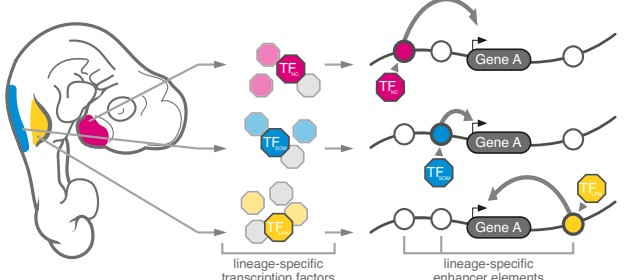

**Fig. 7 | Model for the transcriptional convergence of skeletogenic cells of different embryonic origins.** Depending on anatomical location, different embryonic lineages give rise to the skeletal elements of the vertebrate skeleton. These mesenchymal PC express distinct lineage-specific transcription factor profiles, according to their embryonic origin, which are integrated at lineage-specific enhancer elements to activate a shared set of genes belonging to a core transcriptional program of vertebrate skeletal progenitors.

transcribed in the respective convergence trajectories, yet seem to rely on largely non-overlapping, lineage-specific sets of putative enhancer elements (Figs. 3j–l and 5f). Using in vivo reporter assays, we demonstrate the lineage-specific enhancer activities at a core chondrogenic factor (Fig. 6f, g). Human genetics and molecular studies using in vivo and in vitro models have previously documented the presence of large regulatory landscapes at skeletogenic genes[57–59]. We argue that this regulatory strategy enables distinct up-stream *trans* inputs to activate a common downstream program in the different embryonic lineages, through integration at the level of *cis* elements (Fig. 7). In this scenario, the implementation of distinct long-range enhancer actions would transcriptionally recode distinct PC signatures towards a similar skeletogenic cell fate.

Across the different embryonic lineages, it is unlikely that this convergence involves only a single skeletogenic "master regulator"[5,60]. Indeed, *SOX9*—a prime candidate for such function—acts in many non-skeletogenic lineages[61], and its overexpression is unable to fully reprogram cells with skeletogenic potential toward a chondrocyte fate by itself[62]. Rather, an entire battery of transcriptional regulators—some shared, yet many lineage-specific—seems to drive the lineage-specific convergence toward a skeletogenic phenotype, with further distinctions present at the level of distal *cis* elements (Figs. 2h–j and 3d–l). This apparent regulatory complexity appears further potentiated by differences in transcription factor binding motifs in mesenchymal cells across the embryonic origins (Fig. 4h–j). Whether due to lineage-specific differences in a local chromatin environment or the presence of distinct heteromeric binding partners, such expanded "regulatory grammar" at the level of motif diversity is known to increase the combinatorial flexibility and complexity in cell fate decision and patterning processes[37–39]. Here, our comparative RIME analyses should serve as a valuable resource for future investigations to disentangle the underlying molecular mechanisms shaping lineage-specific DNA binding motif preferences and their functional consequences during tissue maturation and diversification, as well as cell type evolution.

## Origin-specific evolutionary trajectories of skeletogenic cells

The distinct regulatory modalities that specify the skeletogenic cells in different parts of the vertebrate embryo also hold implications for how we treat them in an evolutionary comparative context. In vertebrates, different anatomical regions have acquired the potential to form an endoskeleton at distinct evolutionary time points. It is generally believed that an acellular, cartilage-like support of feeding structures preceded vertebrates[19]. Possible incorporation of the underlying gene regulatory network into a new set of PC cells could then have paved the way for the emergence of the vertebrate cranium[63–65], with additional developmental lineages simply repeatedly co-opting this core

skeletogenic logic to specify a shared cellular phenotype from distinct embryonic sources. Shared expression of pro-skeletogenic factors could thus result from "serial homology" amongst early skeletogenic cells of different embryonic origins[8,10], with the presence of lineage-specific factors simply reflecting "transcriptional noise"–i.e., evolutionary remnants from their respective developmental origins, maintained by stabilizing selection[5,66].

However, the partial regulatory independence we document here, at both *trans* and *cis* levels, implies re-use with substantial network modifications and likely functional implications resulting therefrom. Namely, embryonic lineage-specificities of select transcriptional regulators and enhancer activities appear to be evolutionarily conserved, over hundreds of millions of years, and maintained to later developmental stages of skeletal tissues maturation[25,52,53]. Future studies should thus aim for a refined phylogenetic sampling, as well as investigate later embryonic and post-embryonic dynamics in skeletal cells building the different mature tissue types of the skeleton. Furthermore, in the three embryonic lineages, a distinct regulatory logic may allow for skeletogenic induction to be driven by different extracellular signals–e.g., SHH in somites[67] and Wnt/FGF in the LPM[68]–and for changes in signaling levels to modulate their specification across evolution[6,10]. Last but not least, the possible incorporation of skeletogenic factors into distinct transcriptional complexes would classify the skeletal cells of the three embryonic lineages as independent cell types, according to the "core regulatory complexes" (CoRC) concept[1]. While these different regulatory strategies might originally have been a necessity–to integrate distinct *trans*- and *cis*-regulatory modalities as well as extracellular signaling environments–ultimately, they might have proven beneficial and even adaptive. Namely, the skeletogenic cells of the different embryonic lineages seem to possess distinct character identity mechanisms and, accordingly, have individualized evolutionary trajectories available to them[1,10]. By relying on partially distinct specification networks and, thus, reduced pleiotropy, independent changes in effector gene expression–e.g., to affect extracellular matrix composition and ensuing tissue properties[54,69]–or cellular growth dynamics become possible in the different parts of the skeleton[70–72]. Indeed, naturally occurring genetic variation, as well as induced targeted mutations in some of the embryonic origin-specific regulators we document here, show anatomical location-restricted effects[72–74]. For example, an *ALX1*-containing haplotype has been linked to the diversification of beak shapes and sizes in Darwin's finches[72], while regulatory changes at the *PRXX1* locus appear to contribute to the elongation of forelimb elements in bats[70]. Hence, despite seemingly similar cellular phenotypes, the distinct regulatory strategies at work in the three embryonic lineages may help to make different parts of the vertebrate skeleton become independent targets of evolutionary selection, with distinct biomaterial and patterning properties resulting therefrom.

# Methods

## Tissue collection

Fertilized chicken eggs (Gallus gallus domesticus, "Hubbard") were purchased from local vendors in Switzerland and incubated to the desired stages in a humidified incubator. Embryos were dissected in ice-cold PBS and staged according to Hamburger and Hamilton[75]. Embryonic tissue was either processed for single-cell functional genomics experiments or fixed in 4% PFA at 4 ° C. Embryos of both sexes were included in all analyses. In accordance with Swiss national guidelines (Swiss Animal Protection Ordinance; TSchV, chapter 6, Art. 112), no formal ethics approval was required, as all experiments were carried out before the third trimester of incubation.

## RNA in situ hybridization

Cranial or brachial tissue samples ranging from HH17 to HH24 were dehydrated and cryo-embedded side-by-side in OCT to allow for staining of the entire time series on single slides. Sectioning was performed on a *Leica CM3050S* cryostat, and RNA ISH against *SOX9* and *ACAN* was performed using standard protocols[76]. Brightfield images were acquired on an *Olympus FLUOVIEW FV3000* and globally processed for color balance and brightness using *Adobe Photoshop*.

## Statistics and reproducibility

No statistical method was used to predetermine sample size. No samples were excluded, but low-quality cells were removed from further analyses based on quality control metrics detailed below under "scRNA-seq data pre-processing." The experiments were not randomized, and the investigators were not blinded to allocation during experiments and outcome assessment.

## scRNA-seq data collection

We sampled the frontonasal prominence at stages HH15, HH18, and HH22, the dorsal part of the brachial region at stages HH12, HH15, and HH20, and entire forelimbs at stages HH21, HH24, and HH27[77]. Tissue was dissociated using enzymatic digest (0.25% trypsin in DMEM, for 15 min at 37 °C), with cell capture, cDNA generation, preamplification, and library preparation according to the *10x Genomics Chromium 3′ Kit* instructions and sequencing on *Illumina* platforms[26]. Data was processed and mapped with *CellRanger* (*10x Genomics*), using our in-house improved GRCg6a genome annotation with elongated 3′ UTRs[77].

## scRNA-seq data pre-processing

Unique molecular identifier (UMI) count matrices were filtered for quality based on a cell's total and relative UMI counts (i.e., >4*mean and <0.2*median of the sample), percentage of mitochondrial UMIs (i.e., >median +3*MAD (median absolute deviation) and >0.1, except if UMI count >median). Finally, we calculated UMI count-to-genes detected ratios and removed cells with a ratio <0.15, except if a cell had <2/3 of the max. number of genes detected[77]. In total, 8468 cells remained for frontonasal (2702, 4558, and 1208 cells for HH15, HH18, and HH22), 7777 cells for somite (3093, 2993, and 1691 cells for HH12, HH15, and HH20), and 10,350 cells for forelimb (2987, 5293, and 2070 cells for HH21, HH24, and HH27).

## scRNA-seq data normalization, dimensionality reduction, and clustering

Using the *R* package *Seurat* (v4)[24], UMI counts were normalized by sequencing depth and log transformed. A cell cycle score was calculated using *SCRAN*[78]. Variations of sequencing depth, mitochondrial UMI percentage, and the difference in S and G2M cycle scores were regressed out using *SCTransform* from *Seurat*[24,77,79]. Genes with a higher value of standardized variance than the sum of median and MAD were considered as "highly variable." These steps were carried out independently for the three stages of the three embryonic origins.

Using *Seurat*, we integrated samples from the same embryonic origin and used principal component analysis (PCA) on highly variable genes, followed by tSNE and FFT-accelerated Interpolation-based tSNE algorithms[80] for non-linear dimensionality reduction on the first 19 (nasal), 21 (somite), and 19 (limb) principal components. Using *Seurat* functions, we performed Leiden graph-based clustering[81] on all cells with a resolution of 0.2 (="broad clustering.") A second round of clustering was conducted on select mesenchyme populations, with resolutions of 0.4 (somite, limb) and 0.5 (nasal) ( = "fine clustering.") Cell type assignments of clusters were based on visual inspection of known marker gene expression patterns, and the activity of the two previously identified EC gene expression modules "IMM" and "RED"[25,26] using the *Seurat* function "AddModuleScore."

## Differential expression analysis

Differential expression analysis was based on a logistic regression framework[82] using *Seurat*, with cell cycle differences and embryonic

stages as latent variables. Genes expressed in at least 10% of the cells and showing differences with an adjusted p-value < 0.05 and a log fold change >0.5 ("broad") or >0.25 ("fine") were considered as significantly differentially expressed. To minimize batch effects, differential expression analysis of chondrocytes from different embryonic origins was performed on pseudobulk counts using the *R* package *muscat*[83].

## scRNA-seq data integration across embryonic origins

We filtered out potential doublets using the *R* package *doubletFinder*[84] and removed clusters enriched for mitochondrial counts. The resulting UMI count matrix was divided by size factor and log-transformed using *SCRAN*[78]. The top variable genes (getTopHVGs, *SCRAN*) identified in at least two samples were kept for downstream analyses. Using *Seurat*, we then integrated the count matrices using anchors in canonical correlation analysis (CCA) reduction to compute batch-corrected matrices of the three embryonic origins. To calculate co-embedding projections, the PCA dimension was reduced sample-wide. Anchors for integration were identified using "FindIntegrationAnchors" in reciprocal PCA reductions. We used "IntegrateEmbeddings" to integrate PCA reduction, followed by tSNE calculations ('RunTSNE'). Correlation analyses were performed on "pseudobulk" average gene expression values (*Seurat* function 'AverageExpression') in each cluster.

## scRNA-seq pseudotime analyses

We generated spliced/unspliced count matrices of our selected mesenchymal populations using *velocyto*[85] and assessed the directional transcriptional dynamics of highly variable genes with sufficient spliced/unspliced counts in *scVelo*[27] with the default parameters. We visualized the recovered dynamics on FIt-SNE (fast interpolation-based t-SNE[80]) projections of the three embryonic origins. We then used these Flt-SNE embeddings as input space and constructed a minimum spanning tree with a preset start cluster in the *R* package *slingshot*[30]. In a complementary approach, we integrated samples from the same embryonic origin and used PCA on highly variable genes, followed by force-directed graph drawing[32] for non-linear dimensionality reduction on the first 2 principal components. We then performed tree learning with simplePPT to get the respective pseudotime trajectories using the python package *scFates*[31] (v1.0.1). Alignment of embryonic origin-specific *slingshot* pseudotimes was performed with *TrAGEDy*[33] using 40 interpolated points along the respective chondrogenic trajectories. Module expression dissimilarities were calculated by Spearman correlation (1-ρ), and optimal alignment was identified by dynamic time warping with default settings. Using the *R* package *tradeSeq*[34], we detected temporally differentially expressed genes along the respective chondrogenic trajectories.

## scATAC-seq data collection

Tissue dissociation was performed as previously described[26]. Cell concentration and viability were assessed using the *Nexcelom Cellometer K2*, and ~1*10^6 cells were used to perform nuclei isolation following the *10x Genomics* protocol. Nuclei suspensions were loaded onto a Next GEM chip H, and transposition, nuclei partitioning, and library preparation were performed according to the *10X Genomics* ATAC User Guide. scATAC libraries were quantified on an Agilent 2100 Bioanalyzer system (Agilent) and sequenced on a NovaSeq 6000 system (Illumina).

We used *CellRanger ATAC v1.2.0* (*10x Genomics*) for read processing and quantification and mapped the fragments to the chicken ENSEMBL genome Gallus_gallus-6.0[86] with our in-house improved GRCg6a genome annotation[77]. PEAK_MERGE_DISTANCE was changed to 50, with all other parameters at default settings.

## scATAC-seq data pre-processing

We removed doublets with *ArchR* (v1.0.1)[44] and selected high-quality cells in *Signac* (v1.1.1)[87] using the following thresholds: total number of

fragments in peaks ranging from 1000 to 100,000, fraction of reads in peaks >15%, nucleosome signal <4, and TSS enrichment score >2. Using these criteria, we ended up with 11,527 cells for frontonasal (6171 and 5356 cells for HH15 and HH18), 14,106 cells for somite (11,232 and 2874 cells for HH12 and HH15), and 10,982 cells for forelimb (4453 and 6529 cells for HH21 and HH24).

## scATAC-seq data merging, dimensionality reduction, and clustering

We merged samples from the same embryonic origins and summed fragment counts in 5 kb genomic tiling windows located on autosomes and chromosome Z (208,680 tiles in total). We performed latent semantic indexing (LSI) dimension reduction on a term frequency-inverse document frequency (TF-IDF) normalized matrix with the top 75% of tiles (top 0.1% tiles are removed, putative repetitive elements, or alignment errors) using *Signac* and removed batch effects on LSI components using *Harmony*[88]. We used tSNE and FFT-accelerated Interpolation-based t-SNE algorithm[80] to carry out non-linear dimensionality reduction with LSI dimensions 2:30, and performed Leiden graph-based clustering in *Seurat* on all cells with resolutions of 0.4 (nasal, somite) and 0.6 (limb) (="broad clustering"), and a second round of clustering on select mesenchyme populations with resolution of 0.4 (="fine clustering"). We annotated cell types for both "broad" and "fine" clusters using scATAC-seq gene activity matrices (promoters and gene bodies) and scRNA-seq expression data, with a combination of label transfers in *Seurat* and NNLS regression on cluster-specific genes[89], as well as manual inspection of peaks at known marker genes.

## Peak calling and differential accessibility analysis

We identified peaks using *MACS2* (version 2.2.7.1)[29] with parameters "--nomodel --shift 100 --extsize 200 --keep-dup all --call-summits" on pseudobulks of each cluster, for each embryonic origin, respectively. Peaks used summits as center and were extended to a width of 501 bp. We merged peaks from different clusters of the same embryonic origin and, for overlapping peaks, kept only the most significant one, using adapted code from *ArchR*. To get a consensus peak set, we merged peaks from three embryonic origins and removed redundant and/or overlapping peaks using the same logic.

Differential accessibility analysis was performed in *Seurat* using the total number of fragments and embryonic stages as latent variables. Peaks accessed in at least 10% of the cells and showing differences with adjusted p-value less than 0.05 and a log fold change larger than 0.25 were considered as significantly differentially accessed. Peak-centered heatmaps of differentially accessible peaks were visualized with *deepTools2*[90].

## scATAC-seq data integration across embryonic origins

The top variable peaks (getTopHVGs, *SCRAN*) identified in at least two samples were kept for downstream analyses. To calculate a co-embedding projection, first, we performed reciprocal LSI-dimensional reduction to find anchors (*Seurat* function "FindIntegrationAnchors") and constructed transformation matrices between each query cell and anchor. We computed the integration matrices based on the original LSI matrix with dimensions from 2 to 30 and the transformation matrix using the *Seurat* functions "IntegrateEmbeddings" and "runTSNE" on the integrated LSI dimensions. To remove the batch effects among peak matrices after merging, we binarized the matrix based on the presence/absence of counts. Correlation analyses were performed on "pseudobulk" average count values in each cluster using the function "AverageExpression."

## scATAC-seq pseudotime analyses

We transferred pseudotime values from our scRNA analyses using the *Seurat* function "TransferData." First, we integrated scRNA-seq

expression matrices and scATAC-seq gene activity matrices and performed CCA dimensional reduction to find anchors. We then constructed a transformation matrix between each query cell and each ancho (*Seurat* function "FindTransferAnchors") and computed the transferred scATAC-seq pseudotimes based on the original scRNA-seq pseudotimes and the transformation matrices. For all three embryonic origins, we restricted this transfer to only chondrogenesis-related cell type clusters.

### De novo motif enrichment analysis and annotation

We performed de novo motif enrichment analysis for each cluster, using *Homer*[35] "findMotifsGenome.pl" with -mset vertebrates -size -250, 250 -fdr 5, and motif length between 8–22 bp (*Homer p*-value < 1e-11), using highly accessible peaks for each cluster. We obtained candidate TF annotations for this set of de novo motifs with the help of three databases (*Homer* vertebrates, *JASPAR20* vertebrates[91], *CisBP* v2 chicken[92]) using *Homer* and *STAMP*[93]. We selected the best matches based on scRNA-seq expression levels of the predicted TFs and Spearman correlations between motif activity and gene expression of candidate TFs in scATAC and scRNA aggregates (default $k = 50$, $n = 400$; for small clusters, $k = 20$, $n = 200$). For motifs with similarity scores >0.8, only the one with the lowest p-value was retained. Additionally, we checked for paralog TFs expression along our pseudotime trajectories and calculated its expression correlation to motif activity. We combined annotated de novo motifs from each embryonic origin and calculated motif similarity scores using *PWMEnrich*[94]. For our final set of annotated de novo motifs, we computed per-cell motif deviation scores using *chromVAR*[95] and conducted analysis of differential motif activity using *Seurat*.

### Rapid immunoprecipitation mass spectrometry of endogenous proteins (RIME)

We performed RIME on skeletogenic cells from the three different embryonic lineages using antibodies directed against SOX9 and FOXP1, adapting the general workflow outlined in ref. 40. In total, we analyzed 30 samples (4 replicates per antibody and embryonic origin ($n = 24$), plus two replicates per antibody and two negative controls in our limb test run ($n = 6$)). Briefly, we microdissected tissue from embryos at stage HH29. To avoid contamination from neuronal cells, which express both SOX9 and FOXP1 at this stage, we removed the spinal cord (and all ventral tissues) from our SOM and focused on the first and second pharyngeal arches for our neural crest-derived skeletogenic tissue. We cross-linked the tissue using 2 mM DSG (Disuccinimidyl glutarate) for 40 min and 1% of Formaldehyde (FA) for 13 min at room temperature. After quenching in 125 mM glycine, we extracted, washed, and lysed nuclei using LB1, LB2, and LB3 solutions[40], respectively, and sonicated the material on a Diagenode Bioruptor 300 sonicator 30 s/30 s 20 cycles. The resulting lysates were incubated overnight at 4 °C on a rotating wheel with antibodies against SOX9 (rabbit, Millipore AB5535; 5μg/replicate) or FOXP1 (rabbit, Abcam ab16645; 5 μg/replicate) and magnetic beads (20 μl of Dynabeads ProtA and 20 μl of Dynabeads ProtG per replicate). For our limb test run, we included IgG-only (5μg/replicate) and bead-only controls. Samples were washed seven times with RIPA buffer and two times with AMBIC[40] on beads and processed for LC–MS/MS mass spectrometry analyses. For a detailed description of the mass spectrometry procedure, please refer to Supplementary Data 1, "Sample prep and MS specs." Briefly, proteins were eluted from the magnetic beads, alkylated, and digested using S-Trap™ micro spin columns (Protifi) according to the manufacturer's instructions. Peptides were then eluted, dried, resuspended, and subjected to LC–MS/MS analysis using an orbitrap fusion lumos mass spectrometer fitted with an EASY-nLC 1200 (both Thermo Fisher Scientific). The acquired raw files were searched using MSFragger (v.

4.1) implemented in FragPipe (v. 22.0) against a *Gallus gallus* database (consisting of 43711 protein sequences downloaded from Uniprot on 20231218) and 392 commonly observed contaminants using the default "LFQ-MBR" workflow. Quantitative data was exported from FragPipe and analyzed using the MSstats R package v.4.13.0[96]. Data were imputed using "AFT model-based imputation" and statistics for pairwise comparisons were calculated using the *limma* package[97]. Only DA proteins with a p-value < 0.05 were considered for further analyses.

### Peak-to-gene link analysis

We generated imputed pseudoexpression data for each scATAC cell based on scRNA-seq data, using the *ArchR* function "addGeneIntegrationMatrix." 500 cell aggregates were generated with 100 cells per aggregate. We then computed the Pearson correlation between peak accessibility and pseudoexpression in mesenchymal aggregates using the *ArchR* function "addPeak2GeneLinks." The clustering of peak-to-gene links was calculated by the hkmeans method in *factoextra*[98]. Functional enrichment analyses of peak-to-gene link clusters were conducted in rGREAT[99].

### Evolutionary conservation analysis

We investigated the sequence evolutionary conservation of CREs identified through peak-to-gene link analysis using the phastCons program. Specifically, we retrieved phastCons scores calculated based on multiple alignments of 77 vertebrate species, including 55 birds, from the UCSC Genome Browser website. For each position along the chicken genome, the *phastCons* score represents the probability of negative selection[100]. We then calculated the average *phastCons* scores along the coordinates of CREs that are common between all three origins, CREs shared between two or more origins, and origin-specific CREs.

### Enhancer reporter assays

Genomic regions of candidate enhancers were amplified by PCR from chicken genomic DNA and cloned upstream of a minimal promoter driving GFP expression. Fertilized chicken eggs were incubated at 38.5 °C in a humidified incubator to stage HH14 for lateral plate mesoderm and somite electroporation and stage HH7 for neural crest electroporation. DNA solutions containing our enhancer reporter constructs and a constitutively expressing tdTomato co-electroporation control were injected and EP into epithelial PC populations of the cranial neural crest, the brachial somites, or the lateral plate mesoderm at forelimb levels[101–103]. Embryos were harvested two days post-electroporation, and tissue was processed for immunohistochemistry[104].

### Reporting summary

Further information on research design is available in the Nature Portfolio Reporting Summary linked to this article.

## Data availability

The functional genomics data generated in this study have been deposited in the GEO repository under accession codes GSE281769 (scRNA-seq) and GSE281763 (scATAC-seq)). Previously published samples (limb scRNA-seq stages HH21, 25, and 27[77]) are also available at GEO (accession code: GSE174565). The proteomics data generated in this study have been deposited to the ProteomeXchange Consortium with identifier PXD057934 via the MassIVE partner repository with MassIVE data set identifier MSV000096424.

## Code availability

We used previously published computational tools, with specific settings detailed under https://github.com/wangmhan/skeletoConvergence[105].

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

## Acknowledgements

The authors wish to thank I. Adameyko and all members of the lab for insightful discussions, R. Sheth for advice on RIME experiments, G. Viktorin for help with embryo collections for RIME experiments, as well as the joint "Genomics Facility Basel" at ETHZ D-BSSE and the "Proteomics Core Facility" at Biozentrum, University of Basel, for help with functional genomics and proteomics experiments, respectively. Calculations for single-cell functional genomics analyses were performed at sciCORE (http://scicore.unibas.ch/), scientific computing center at the University of Basel. This work was supported by research funds from the Swiss National Science Foundation [SNSF project grant number 310030_189242 to P.T.], the Swiss 3R Competence Centre [3RCC grant OC-2018-005 to P.T.] and the University of Basel to P.T.

## Author contributions

This study was conceived and designed by M.W., A.D.P.T., C.F., M.L. and P.T. Single-cell functional genomics data was generated by C.F., M.L., C.M. and A.F. Bioinformatics analyses were conducted by M.W., C.F., A.F., and P.T. RIME experiments were performed by M.L. and analyzed by D.R. and P.T. Enhancer reporter assays were performed by A.D.P.T. and S.F. P.T. wrote the paper, with feedback from all other authors.

## Competing interests

The authors declare no competing interest.
