## [Transparent Peer Review file · Nature Communications]

Distinct Gene Regulatory Dynamics Drive Skeletogenic Cell Fate Convergence During Vertebrate Embryogenesis

Corresponding Author: Professor Patrick Tschopp

Version 0:

Reviewer comments:

Reviewer #1

(Remarks to the Author)

In this paper the authors provide evidence that phenotypically identical skeletogenic cells in the chicken skeleton from different embryonic origins are controlled by embryonic origin specific trans-regulatory programs. These three embryonic origins correspond to the three independent evolutionary origins of skeletal elements, the cranial, axial and appendicular skeleton. This conclusion is supported by multiple lines of evidence including RNAseq profiles along differentiation trajectories, chromatin accessibility profiles, transcription factor binding motif profiles, the location of lineage specific cis-regulatory elements (proximal promoters cell function specific, distal CRE lineage specific), and CRE to gene linkage; as well as reporter gene activity driven by cell type specific enhancers. In sum these lines of evidence convincingly show that skeletogenic cells represent three different cell types (or cell type clades?) with convergent phenotype (chondrocyte phenotype). Importantly, cell type identity is linked to different transcription factor binding motives of common skeletogenic transcription factors, suggesting the existence of cell type specific transcription factor complexes, call Core Regulatory Complexes, as predicted in their reference (1). Finally the authors suggest the important evolutionary hypothesis, namely the potential independent evolutionary potential of these three parts of the vertebrate endoskeleton. This is an exciting and paradigm shifting paper that has a high potential of becoming a classic in developmental evolutionary biology.

Minor comment:

It is exciting to see the lineage/cell type specific transcription factor binding motifs suggesting the activation of different transcription factor complexes in these three cell types. It might be possible to predict the other members of the transcription factor complex associated with say SOX9 in different cell types. Do these profiles contain enough information to infer these other transcription factors?

One should encourage readers to investigate the conservation of these trans-regulatory programs in other vertebrates, say amphibians and mammals. It would be interesting to predict that these cell type specific programs are conserved, even though testing that is clearly beyond the scope of the current submission. But planting the seed for other researchers to test this prediction would enhance the impact of this paper.

(Remarks on code availability)

My reading suggests that all analysis is based on published computational tools and resources.

Reviewer #2

(Remarks to the Author)

The study is focusing particularly on the development of skeletogenic cells in vertebrates which are crucial for the development of their endoskeletons. These cells originate from three different embryonic precursor lineages: the neural crest, the somites, and the lateral plate mesoderm. Despite their different origins, these cells converge towards similar skeletogenic phenotypes during development. This convergence has evolved at different times across the lineages, raising questions about the homology of the vertebrate skeleton's various parts and genetic programs. The study investigates how molecular properties specific to each lineage are integrated at the gene regulatory level to achieve this phenotypic

convergence. Using techniques like single-cell transcriptomics and chromatin accessibility profiling, the authors examine the gene regulatory dynamics that enable these cells to converge towards a skeletogenic fate. They find that distinct transcription factor profiles inherited from each precursor state, along with lineage-specific enhancer elements, play crucial roles in executing a core skeletogenic program.

I think this is a timely and technically sound manuscript, though I have several comment that must be address before publication (some of them are more philosophical and some are technical):

1. The authors mentioned homology and they need to be more clear what type of homology they mean: the homology of a gene expression program (sets of regulated pro-skeletogenic genes – presumably homologous), a homology of regulation of these gene expression programs (genomic regulatory landscape – presumably non-homologous in this case, evolved independently or?), the homology of skeletogenic cell types, integrated tissue structures or something else. The definitions need to be very clear and the discussion part shall be improved by systematically going through these levels of homology to bridge all findings in this light.
2. It is exceptionally hard to reliably and stably navigate trajectory in blobby transcriptional or ATACseq objects (actually, all embeddings in this study are blobby and do not form slender transitions) in a multidimensional space, and therefore I would like the authors to use alternative methods of clustering and trajectory navigation for obtaining the consistency of the analysis with a different approach. For this, I suggest scFates as a tool (or other tools) to find alternative trajectories with a different method. Showing the consistency of the current method and some other methods that solve the same task in different ways will make this story more technically sound. Also, navigating trajectories in transcriptional data, not only ATACseq data will be good for showing that the same genes (as in ATAC trajectory) are being transcribed in a similar sequence – another good cross-validation.
3. I would like to ask for a supplementary figure, where the clusters in Figures 1, 2, 3 will be explained in terms of specific or differentially expressed markers. For example, what is the basis for claiming that some cell are skeletogenic? I trust that the authors checked all of that and have a good answer before proceeding to downstream analysis, but that must be clearly revealed in supplementary technical figures (heatmaps, dotplots showing top 20 genes defining each cluster etc.).
4. The authors must provide currently missing quality control figures showing the coverage, proportion of mitochondrial genes, doublet removal for every dataset.

(Remarks on code availability)

Reviewer #3

(Remarks to the Author)

General Comments:

The manuscript presents an intriguing investigation into the gene regulatory mechanisms underlying the skeletogenic potential of mesenchymal precursors from various embryonic origins. Considering this from an evolutionary perspective adds particular interest and value to the study. The authors employ scRNA-Seq and scATAC-seq to address two key questions: How are lineage-specific molecular properties of the three precursor pools integrated at the gene regulatory level to allow for phenotypic convergence towards a skeletogenic cell fate? And to what extent can the convergently specified cells of the vertebrate skeleton be considered truly homologous?

The study provides a valuable overview of the data, but it lacks detailed mechanistic insights. Without these details, the manuscript serves only as a useful resource for further studies.

Major Comments:

1. It would be beneficial for the manuscript to clearly delineate the underlying hypotheses and explore potential alternative explanations. By considering alternative hypotheses, the study's findings can be contextualized to reveal their true significance and unexpected insights, enriching the overall impact of the research
2. A review of the literature on different regulatory strategies and an assessment of homology among cell types would enrich the manuscript. Integrating these discussions would demonstrate a thorough understanding of the field and how the current study contributes to existing knowledge.
3. The study lacks detailed mechanistic insights. One possible direction to pursue (though not the only one) is a detailed delineation of the transcription factors (TFs) and their binding sites, likely to initiate Sox9 expression in each mesenchymal lineage. Additionally, identifying whether there is a switch in TFs and/or binding sites that drive Sox9 expression along the differentiation continuum of chondrocytes would be of interest (from undifferentiated mesenchymal cells to chondrocytes). Further, investigating whether there is a hierarchical organization of TFs, for example, are where primary TFs initiate core chondrogenic genes and secondary TFs drive lineage-specific genes? This would help clarify the sequential and coordinated regulation of gene expression during chondrogenesis.
4. The discussion section primarily summarizes the findings without providing sufficient interpretation. Exploring potential mechanisms underlying the observed differences in regulatory strategies among skeletogenic cells and discussing their implications, including their relevance to endochondral vs intermembranous ossification processes (for example), is essential. Additionally, addressing potential study limitations and proposing future research directions based on the generated data would enhance the manuscript's relevance and impact.

(Remarks on code availability)

Reviewer #4

(Remarks to the Author)

In this manuscript, the authors address the question of how distinct embryonic lineages of cells—the cranial neural crest, the somitic sclerotome, and the somatopleure of the lateral plate mesoderm—differentiate into a skeletogenic cell fate. They use a chick as a model and conduct an in-depth analysis of scRNA and scATAC-seq data. From the scRNA-seq data, they find a transcriptional convergence from three discrete embryonic lineages, though TF-specific fine mapping revealed a lineage-specific repertoire of expressed transcription factors. This suggests that distinct upstream transcriptional factors regulate skeletal lineage differentiation. Consistent with this, from the scATAC-seq data analysis, they find distinct putative distal enhancers regulating this cell fate decision in each anatomical lineage and identify enriched TF motifs in each lineage. Lastly, through the combination of scRNA and scATAC-seq, they find SOX9 enhancers in each anatomical lineage.

This biological question is intriguing, and the data showing different trans- and cis-regulatory elements regulating this cell fate is a valuable resource. I also have the impression of re-evaluation of some already known lineage-specific TFs by single-cell analysis, though it is still a great resource and worth publishing in Nature Communications.

I have some comments below:

1. First of all, I do not see any data availability section. The authors must deposit their data in a public repository such as GEO (<https://www.ncbi.nlm.nih.gov/geo/>). They can provide a reviewer access token, and data under review is secured. Upon revision, authors must submit their data and make it available to the public after publication.
2. The authors emphasize the evolutionary aspects, but I don't see significantly impressive data for this aspect. Authors can analyze the conservation of distinct enhancers in each lineage and other overlaps with evolutionarily important elements such as chicken accelerated regions (CARs) and avian-specific highly conserved elements (ASHCEs) (e.g., PMID: 34584102 and PMID: 28165450).
3. Chick limb buds bulk RNA-seq/ATAC-seq analysis has been performed by Dr. Rolf Zeller's group (PMID: 34584102). The authors should discuss what new insights are provided compared to previous bulk-seq analysis.
4. SOX9 enhancers have been intensively analyzed in humans and mice. Are the enhancers that the authors found homologous sequences of these previously identified enhancers? Such as PMID: 32991838 and PMID: 25940622. I hope the authors search and refer to previous publications.

Minor comments:

5. For the in vivo enhancer assay data, please show the whole embryo pictures that are more convincing to see the tissue specificity of enhancers.
6. For non-chick researchers, HH15-27 is hard to imagine. Any picture or illustration of embryos would be a great help to understand your manuscript.

(Remarks on code availability)

I couldn't find the code or data availability section. It was impossible to review this.

Version 1:

Reviewer comments:

Reviewer #1

(Remarks to the Author)

This reviewer thanks the authors for their response to my [few] suggestions. This is a great paper with great potential to become a classic in evolutionary developmental biology.

Reviewer #2

(Remarks to the Author)

I am happy with the revision of the manuscript. It is ready to be published.

Reviewer #3

(Remarks to the Author)

The authors have adequately addressed all my comments. I fully support the publication of this manuscript.

Reviewer #4

(Remarks to the Author)

The authors have addressed the reviewers' comments, and the manuscript is now suitable for publication.

REVIEWER COMMENTS

Reviewer #1 (Remarks to the Author):

In this paper the authors provide evidence that phenotypically identical skeletogenic cells in the chicken skeleton from different embryonic origins are controlled by embryonic origin specific trans-regulatory programs. These three embryonic origins correspond to the three independent evolutionary origins of skeletal elements, the cranial, axial and appendicular skeleton. This conclusion is supported by multiple lines of evidence including RNAseq profiles along differentiation trajectories, chromatin accessibility profiles, transcription factor binding motif profiles, the location of lineage specific cis-regulatory elements (proximal promoters cell function specific, distal CRE lineage specific), and CRE to gene linkage; as well as reporter gene activity driven by cell type specific enhancers. In sum these lines of evidence convincingly show that skeletogenic cells represent three different cell types (or cell type clades?) with convergent phenotype (chondrocyte phenotype). Importantly, cell type identity is linked to different transcription factor binding motives of common skeletogenic transcription factors, suggesting the existence of cell type specific transcription factor complexes, call Core Regulatory Complexes, as predicted in their reference (1). Finally the authors suggest the important evolutionary hypothesis, namely the potential independent evolutionary potential of these three parts of the vertebrate endoskeleton. This is an exciting and paradigm shifting paper that has a high potential of becoming a classic in developmental evolutionary biology.

We thank the reviewer for the very positive evaluation of our work. It is obviously rewarding to see this level of enthusiasm, and it only will further motivate us to continue this line of research!

Minor comment:

It is exciting to see the lineage/cell type specific transcription factor binding motifs suggesting the activation of different transcription factor complexes in these three cell types. It might be possible to predict the other members of the transcription factor complex associated with say SOX9 in different cell types. Do these profiles contain enough information to infer these other transcription factors?

We were equally intrigued by the potential of lineage-specific transcription factor complexes, as suggested by these differences in DNA binding motifs, and what they would imply about cell type homology. We thus decided to address this experimentally in a more direct and systematic manner, using RIME (Rapid Immunoprecipitation Mass spectrometry of Endogenous proteins) on two of our identified core

chondrogenic factors, SOX9 and FOXP1. In Fig. 4m (and Suppl. Fig. 7p, plus QC), we now document the likely presence of lineage-specific interaction partners for the two transcription factors, which – in turn – imply distinct core regulatory complexes (CoRCs). This would further substantiate our claim that skeletogenic cells from the three embryonic lineages should be considered as distinct, ‘non-homologous’ cell types.

We would like to stress, however, that we do not believe that this data by itself provides ‘definite proof’ for lineage-specific CoRCs, and hope to have phrased our conclusions resulting therefrom with the appropriate caution in our revised manuscript. Yet, we hope that this reviewer as well as others will appreciate its complementary value, the resource it provides for further experimental investigations, and the potential of RIME in general, in generating orthogonal molecular data to help assess cell type homologies in other contexts, beyond the skeletal system.

In a more targeted manner, and following the reviewer’s suggestion, we also performed a ‘split motif’ analysis on our nasal SOX9 motif. Naturally, using only half of the predicted motif length reduces the information content and indeed we do lose statistical power in detecting significant correlations between motif activities and the corresponding transcription factor expression profiles. Yet, using our original approach on only the 5’ half of our nasal motif, within the Top5 predicted motif hits we obtained, three of them were linked to TEAD transcription factors (2x TEAD3, 1x TEAD1). Intriguingly, the protein interaction of TEAD3 with SOX9 appears specifically enriched in our nasal RIME samples (see below), thus making it a prime candidate for a SOX9 co-binding partner occupying this motif. A more definite answer regarding this question, however, would require further and more directed follow-up experiments, which – we believe – are beyond the scope of this manuscript.

One should encourage readers to investigate the conservation of these trans-regulatory programs in other vertebrates, say amphibians and mammals. It would be interesting to predict that these cell type specific programs are conserved, even though testing that is clearly beyond the scope of the current submission. But planting the seed for other researchers to test this prediction would enhance the impact of this paper.

We do agree with the reviewer about the importance to investigate the functional implications of these distinct regulatory programs in a comparative context. In fact, we

are currently gearing up to do this at both macro- and microevolutionary levels, across the vertebrate clade. Furthermore, we refer now in several instances in the manuscript to related work that has previously been performed in mammals and birds, most notably in the up-dated discussion, lines 533-538 and 555-557.

Reviewer #1 (Remarks on code availability):

My reading suggests that all analysis is based on published computational tools and resources.

This is correct. We do, however, detail the specific parameters used for these computation tools in a GitHub page dedicated to the analyses of this manuscript (see <https://github.com/wangmhan/skeletoConvergence>).

We thank the reviewer for their constructive feedback on our work!

Reviewer #2 (Remarks to the Author):

The study is focusing particularly on the development of skeletogenic cells in vertebrates which are crucial for the development of their endoskeletons. These cells originate from three different embryonic precursor lineages: the neural crest, the somites, and the lateral plate mesoderm. Despite their different origins, these cells converge towards similar skeletogenic phenotypes during development. This convergence has evolved at different times across the lineages, raising questions about the homology of the vertebrate skeleton's various parts and genetic programs. The study investigates how molecular properties specific to each lineage are integrated at the gene regulatory level to achieve this phenotypic convergence. Using techniques like single-cell transcriptomics and chromatin accessibility profiling, the authors examine the gene regulatory dynamics that enable these cells to converge towards a skeletogenic fate. They find that distinct transcription factor profiles inherited from each precursor state, along with lineage-specific enhancer elements, play crucial roles in executing a core skeletogenic program.

I think this is a timely and technically sound manuscript, though I have several comment that must be address before publication (some of them are more philosophical and some are technical):

We thank the reviewer for their assessment of our work as being ‘timely and technically sound’, and we very much appreciate the added value the suggestions of the reviewer have brought to our manuscript (see details below)

1. The authors mentioned homology and they need to be more clear what type of homology they mean: the homology of a gene expression program (sets of regulated pro-skeletogenic genes – presumably homologous), a homology of regulation of these gene expression programs (genomic regulatory landscape – presumably non-homologous in this case, evolved independently or?), the homology of skeletogenic cell types, integrated tissue structures or something else. The definitions need to be very clear and the discussion part shall be improved by systematically going through these levels of homology to bridge all findings in this light.

We are primarily interested in the homology (or homoplasy) argument at the level of the cell type. The re-use of similar sets of pro-skeletogenic genes could indeed imply ‘serial homology’ amongst the skeletal cells of the different embryonic origins (see e.g. argument by DiFrisco and colleagues, Seminars in Cell & Developmental Biology 2022). However, as we now expand upon in the revised introduction, there are different scenarios that could result in similar transcriptional signatures, such as e.g. (evolutionary) convergence. The fact that we see distinct *cis*- and *trans*-regulatory dynamics underlying this (developmental) convergence (see Figs. 2,3,5), and the likely differences in the regulatory protein complexes in which the core pro-skeletogenic genes seem to exert their functions (see Fig. 4), suggests that these cells are in fact non-homologous, and evolutionarily independent. We now explore this also in more detail in our revised discussion.

We thank this reviewer (and reviewer 3) for having brought to our attention that one of the main motivations for our study – i.e., investigating the potential homology versus homoplasy of skeletal cells – had somehow almost gotten lost in the writing of the first version of our manuscript. We hope to have rectified this now with our revised versions of both the introduction and discussion.

2. It is exceptionally hard to reliably and stably navigate trajectory in blobby transcriptional or ATACseq objects (actually, all embeddings in this study are blobby and do not form slender transitions) in a multidimensional space, and therefore I would like the authors to use alternative methods of clustering and trajectory navigation for obtaining the consistency of the analysis with a different approach. For this, I suggest scFates as a tool (or other tools) to find alternative trajectories with a different method. Showing the consistency of the current method and some other methods that solve the same task in different ways will make this story more technically sound. Also, navigating trajectories in transcriptional data, not only ATACseq data will be good for

showing that the same genes (as in ATAC trajectory) are being transcribed in a similar sequence – another good cross-validation.

We thank the reviewer for this suggestion. We have now performed additional ‘scFates’ analyses on our transcriptomic data, and projected the resulting pseudotemporal trajectories on ‘ForceAtlas2’ layouts, for improved cluster resolution (see new Supplementary Fig. 4a-c). The resulting trajectories largely align with our previous ‘slingshot’ analyses (Fig. 3) and ‘scVelo’ (another Pseudotime approach we had already used, but remained somewhat “hidden” in our original Fig. 1i-k): they all reveal a trajectory connecting a “naïve” mesenchymal precursor population to early skeletogenic cells (see also response to point 3).

However, the new ‘scFates’/‘ForceAtlas2’ combo now additionally allowed us to a) visualize the effects of cell proliferation on these projections, even after cell cycle regression, and b), more importantly, clearly delineate the – although expected, but so far not described – independent trajectory of directly/intramembranously ossifying cells in our nasal sample (Supp. Fig. 4a). We decided to still display the ‘scFates’/‘ForceAtlas2’ combo in the supplement only, not to ‘mislead’ the reader about the degree of differentiation of the cells we are studying: although transcriptionally clearly distinct (see response to point 3), these are still early skeletogenic cells.

Additionally, following the reviewer’s suggestion – and combining it with our response to point 3 of reviewer 3 – we now show pseudotemporal trajectories of both motif activities and their corresponding transcription factors’ expression profiles, along the skeletogenic transition in a new Fig. 6c, d. The two modalities show largely congruent activation patterns, thus providing us with an additional cross-validation for the predicted dynamics.

3. I would like to ask for a supplementary figure, where the clusters in Figures 1, 2, 3 will be explained in terms of specific or differentially expressed markers. For example, what is the basis for claiming that some cell are skeletogenic? I trust that the authors checked all of that and have a good answer before proceeding to downstream analysis, but that must be clearly revealed in supplementary technical figures (heatmaps, dotplots showing top 20 genes defining each cluster etc.).

In Supplementary Fig. 2g-h, we now provide an extended list of marker genes that motivated us to describe certain clusters as being ‘skeletogenic’. Furthermore, we would like to highlight the important role that the chondrogenic modules IMM and RED already had in this original assignment: the expression profiles of more than 50 genes are represented in these module activity plots (gene numbers are now indicated in Fig.1h, for clarity), thus minimizing some of the (still) inherent signal-to-noise issues of scRNA-seq, and allowing us to make high confidence predictions.

However, thanks to the reviewer's comment, we also re-visited additional markers for the other clusters we've identified: this allowed us to verify the identity of the "naïve" mesenchymal precursor populations, assess the general proliferative status of our cells, and – importantly – identify nasal 'fine' cluster 5 as "intramembranously ossifying cells" (e.g. expression of SPARC, POSTN) and limb 'fine' cluster 5 as "late chondrocytes" (e.g. expression of RUNX2, FGFR3) (see Supplementary Fig. 2g-h)

4. The authors must provide currently missing quality control figures showing the coverage, proportion of mitochondrial genes, doublet removal for every dataset.

We now document our scRNA-seq quality control pipeline in Supplementary Fig. 1d, with the resulting overall cell numbers and sample-by-sample quality metrics shown in Supplementary Fig. 1e-h. Furthermore, we now also show sample-by-sample quality metrics for our scATAC-seq data in Supplementary Fig. 3a-c.

We thank the reviewer for their constructive feedback on our work!

Reviewer #3 (Remarks to the Author):

General Comments:

The manuscript presents an intriguing investigation into the gene regulatory mechanisms underlying the skeletogenic potential of mesenchymal precursors from various embryonic origins. Considering this from an evolutionary perspective adds particular interest and value to the study. The authors employ scRNA-Seq and scATAC-seq to address two key questions: How are lineage-specific molecular properties of the three precursor pools integrated at the gene regulatory level to allow for phenotypic convergence towards a skeletogenic cell fate? And to what extent can the convergently specified cells of the vertebrate skeleton be considered truly homologous?

The study provides a valuable overview of the data, but it lacks detailed mechanistic insights. Without these details, the manuscript serves only as a useful resource for further studies.

We thank the reviewer for their feedback. We hope that – after addressing their and the other reviewers' comments – our current manuscript now offers more than 'just' a resource, however useful as this may already be. In particular, we have tried to highlight the conceptual aspects concerning 'cell type evolution' and clarify the 'cell

type homology' discussion. We also provide additional data and analyses, as outlined below.

Major Comments:

1. It would be beneficial for the manuscript to clearly delineate the underlying hypotheses and explore potential alternative explanations. By considering alternative hypotheses, the study's findings can be contextualized to reveal their true significance and unexpected insights, enriching the overall impact of the research

We thank this reviewer (and reviewer 2) for having brought to our attention that one of the main motivations for our study – i.e., investigating the potential homology versus homoplasmy of skeletal cells – had somehow almost gotten lost in the writing of the first version of our manuscript.

We have now substantially changed both introduction and discussion, to clarify our reasoning why we think that skeletal cells of the different embryonic lineages should in fact be considered as distinct, non-homologous cell types. Further, we stress the importance of distinct regulatory strategies and transcription factor complexes in this assessment, and explain why we think an alternative scenario – i.e., maintenance of “non-functional transcriptional noise”, due to weak (or lack of) purifying selection on the lineage-specific transcriptomes (lines 526-530) – is counterbalanced by the fact that some of the regulators identified by our analyses as ‘lineage-specific’ seem to have experienced positive selection in naturally occurring populations, to drive species- and anatomical location-specific phenotypes (lines 550-557).

2. A review of the literature on different regulatory strategies and an assessment of homology among cell types would enrich the manuscript. Integrating these discussions would demonstrate a thorough understanding of the field and how the current study contributes to existing knowledge.

As outlined above, we now devote dedicated sections of both introduction and discussion to this important topic. Besides the implications of the distinct *cis*- and *trans*-regulatory dynamics we document – prior to the activation of core pro-skeletogenic factors – we discuss why we think these factors do not work in isolation, but continue to have their activities modulated in a lineage-specific manner. For one, while prior experimental evidence already had suggested that skeletogenic induction relies on shared and essential activators, it is unlikely that the process is driven exclusively by a single ‘master regulator’, working in isolation. Rather, we believe – as it is increasingly being acknowledged in other developmental contexts – that multiple factors are required. These transcriptional regulators can have their DNA binding motifs (and thus regulatory targets) modulated by different co-regulatory binding

partners, in distinct transcription factor complexes. While this had already been suggested by the overall lower similarities of our mesenchymal TF motifs (Fig. 4h-j), with the newly generated RIME data in our revised manuscript we now provide additional experimental data to support this notion (Fig. 4k-m, Supp. Fig. 7). Since we do not believe, however, that our RIME data in itself ‘demonstrates’ lineage-specific CoRCs, we hope to have phrased our tentative conclusions resulting therefrom with appropriate caution. Yet, we do believe that this data set represents a valuable resource for further investigations, and hope that RIME, or other proteomics-based approaches, may add an additional layer of regulatory complexity in the assessment of cell type homologies, for the vertebrate skeleton and beyond.

3. The study lacks detailed mechanistic insights. One possible direction to pursue (though not the only one) is a detailed delineation of the transcription factors (TFs) and their binding sites, likely to initiate Sox9 expression in each mesenchymal lineage. Additionally, identifying whether there is a switch in TFs and/or binding sites that drive Sox9 expression along the differentiation continuum of chondrocytes would be of interest (from undifferentiated mesenchymal cells to chondrocytes). Further, investigating whether there is a hierarchical organization of TFs, for example, are where primary TFs initiate core chondrogenic genes and secondary TFs drive lineage-specific genes? This would help clarify the sequential and coordinated regulation of gene expression during chondrogenesis.

We thank the reviewer for this excellent suggestion. Using our custom position weight matrices library, we first set out to search for TF motifs that were over-represented in a lineage-specific manner, in differentially accessible peaks (DAPs) of skeletogenic cells within a 1 Megabase interval centred on the SOX9 locus. We display their relative enrichment in panel b of our new Figure 6. Then, using the same TFs, we display their motif activities and RNA expression profiles, along the respective pseudotime trajectories of the embryonic origins their motifs were found to be enriched in (panels c, d). On the one hand, the (largely) overlapping dynamics, between motif activity and TF expression, corroborate our motif assignments (see also comment 2, reviewer 2). On other hand, it suggests a stepwise and lineage-specific activation of SOX9, and thereby identifies candidate genes in controlling these key dynamics. Lastly, we would like to point out that our RIME experiments hint at an additional layer of lineage-specific regulatory complexity, after the initial activation of some of the shared core skeletogenic factors – namely, to potentially modulate DNA binding patterns of skeletogenic factors, and, consequently, alter tissue properties and growth dynamics in the different parts of the vertebrate skeleton.

4. The discussion section primarily summarizes the findings without providing sufficient interpretation. Exploring potential mechanisms underlying the observed

differences in regulatory strategies among skeletogenic cells and discussing their implications, including their relevance to endochondral vs intermembranous ossification processes (for example), is essential. Additionally, addressing potential study limitations and proposing future research directions based on the generated data would enhance the manuscript's relevance and impact.

Following this reviewer's suggestion, we have substantially shortened the descriptive part of our discussion and – in line with advice from other reviewers – have expanded the more conceptual aspects concerning cell type evolution and homology assessment, and how they relate to the different levels of regulatory complexity we have uncovered in our study. Also, throughout the revised discussion, we hint at study limitations and putative future directions, for which our data will hopefully have contributed a foundation for.

We thank the reviewer for their constructive feedback on our work!

Reviewer #4 (Remarks to the Author):

In this manuscript, the authors address the question of how distinct embryonic lineages of cells—the cranial neural crest, the somitic sclerotome, and the somatopleure of the lateral plate mesoderm—differentiate into a skeletogenic cell fate. They use a chick as a model and conduct an in-depth analysis of scRNA and scATAC-seq data. From the scRNA-seq data, they find a transcriptional convergence from three discrete embryonic lineages, though TF-specific fine mapping revealed a lineage-specific repertoire of expressed transcription factors. This suggests that distinct upstream transcriptional factors regulate skeletal lineage differentiation. Consistent with this, from the scATAC-seq data analysis, they find distinct putative distal enhancers regulating this cell fate decision in each anatomical lineage and identify enriched TF motifs in each lineage. Lastly, through the combination of scRNA and scATAC-seq, they find SOX9 enhancers in each anatomical lineage.

This biological question is intriguing, and the data showing different trans- and cis-regulatory elements regulating this cell fate is a valuable resource. I also have the impression of re-evaluation of some already known lineage-specific TFs by single-cell analysis, though it is still a great resource and worth publishing in Nature Communications.

We thank the reviewer for their valuable feedback. While we agree that familiar lineage-specific TFs (re-)appeared in our study (and served as an 'internal QC' for us), we also identified many novel candidate factors in our convergence analyses.

Furthermore, the transcriptional dynamics, as the different precursors switch to a skeletogenic fate, plus the diversity in putative enhancers and TF binding motifs uncovered, in our eyes clearly has brought added value, even for some of the 'familiar faces' amongst the identified TFs. More importantly, however, in our revised versions of introduction and discussion, we have tried to stress what these lineage-specific regulatory modalities imply for the evolutionary dynamics in the resulting cell and tissue types, as well as for cell type homology assessments.

I have some comments below:

1. First of all, I do not see any data availability section. The authors must deposit their data in a public repository such as GEO (<https://www.ncbi.nlm.nih.gov/geo/>). They can provide a reviewer access token, and data under review is secured. Upon revision, authors must submit their data and make it available to the public after publication.

The functional genomics data has now been deposited at GEO (GSE174565, GSE281763, GSE281769), and has been made available to the public at the time of re-submission. Furthermore, the new proteomics data has been made available at the ProteomeXchange Consortium (<https://www.proteomexchange.org/>, MassIVE data set identifier MSV000096424 and ProteomeXchange identifier PXD057934).

2. The authors emphasize the evolutionary aspects, but I don't see significantly impressive data for this aspect. Authors can analyze the conservation of distinct enhancers in each lineage and other overlaps with evolutionarily important elements such as chicken accelerated regions (CARs) and avian-specific highly conserved elements (ASHCEs) (e.g., PMID: 34584102 and PMID: 28165450).

We thank the reviewer for this suggestion. We have now intersected our putative enhancer sequences from our peak-to-gene link analyses with the two suggested collections of avian- and chick-specific genomic elements. While we find ASHCEs in more than 15% of our enhancer candidates, we only find 19 CARs in a total of 20'709 CREs. This could imply that these enhancers indeed have experienced avian-specific selective forces, potentially to modulate tissue-specific properties of the bird skeleton. Furthermore, we have performed *phastCons* conservation analyses on our elements. Intriguingly, putative enhancer elements shared amongst embryonic origins showed on average higher conservation scores, indicating that they might have experienced decreased rates of sequence evolution, due to pleiotropic constraints.

3. Chick limb buds bulk RNA-seq/ATAC-seq analysis has been performed by Dr. Rolf

Zeller's group (PMID: 34584102). The authors should discuss what new insights are provided compared to previous bulk-seq analysis.

This important study performed a comparative analysis of the early patterning and signalling feedback dynamics in the emerging limbs of chicken and mice. As such, however, it did not include a cell lineage-resolved analysis of the gene regulatory dynamics of cells undergoing a switch to a skeletal cell fate, nor did it provide a comparison between the three distinct embryonic lineages that form the tetrapod skeleton. However, in our revised discussion we have tried to better contextualize our results in the light of this and other previous studies, which have provided foundational insights into the 'early' as well as 'late' stages of development.

4. SOX9 enhancers have been intensively analyzed in humans and mice. Are the enhancers that the authors found homologous sequences of these previously identified enhancers? Such as PMID: 32991838 and PMID: 25940622. I hope the authors search and refer to previous publications.

None of our enhancers had so far been tested in a skeletogenic context. We only found experimental evidence for homologous enhancer testing for one of our sequences, the nasal one, but in ES cell-based assays using STARR-seq readouts (PMID: 30033119, PMID: 32912294). This could potentially also explain its more 'promiscuous' activity. We now refer to the relevant publications in our revised text.

Minor comments:

5. For the in vivo enhancer assay data, please show the whole embryo pictures that are more convincing to see the tissue specificity of enhancers.

We now provide a spatial and tissue-level context for the observed enhancer activities on zoomed-out tissue sections in our new Supplementary Figure 9. A true 'whole embryo' overview was unfortunately not feasible, due to low contrast resulting from high autofluorescence in the green channel and overall low expression levels of GFP. However, we agree with the reviewer that a tissue-level overview is essential to assess the specificity of our candidate enhancers' signals, which we hope to have provided now with these additional images.

6. For non-chick researchers, HH15-27 is hard to imagine. Any picture or illustration of embryos would be a great help to understand your manuscript.

We thank the reviewer for this suggestion. We now provide embryo drawings of the stages sampled, to provide a morphological and temporal context. We agree with the reviewer's hope that this should make the manuscript more accessible to a general readership.

Reviewer #4 (Remarks on code availability):

I couldn't find the code or data availability section. It was impossible to review this.

We apologize for this confusion. The blame for this falls entirely on me, based on two misunderstandings at the time I was submitting the manuscript. During the submission process, I simply clicked on the 'CodeShare' option for code assessment, having forgotten that our first author, had in fact already prepared a GitHub page detailing the specifics of our bioinformatics analyses. We tried to communicate this error post-submission, but apparently have failed to do so successfully.

However, as Reviewer 1 correctly pointed out in their assessment, we have not produced novel code for the analyses, but rather relied on a variety of previously published computational methods, as detailed in our Material & Methods section. Nevertheless, we believe that having access to the different parameters we used will increase the accessibility and reproducibility of our results. The reviewers and readers can access them now under <https://github.com/wangmhan/skeletoConvergence>.

We thank the reviewer for their constructive feedback on our work!